# LOCALIZED GRAPH CONTRASTIVE LEARNING

## ABSTRACT

Contrastive learning methods based on InfoNCE loss are popular in node representation learning tasks on graph-structured data. However, its reliance on data augmentation and its quadratic computational complexity might lead to inconsistency and inefficiency problems. To mitigate these limitations, in this paper, we introduce a simple yet effective contrastive model named Localized Graph Contrastive Learning (LOCAL-GCL in short). LOCAL-GCL consists of two key designs: 1) We fabricate the positive examples for each node directly using its first-order neighbors, which frees our method from the reliance on carefully-designed graph augmentations; 2) To improve the efficiency of contrastive learning on graphs, we devise a kernelized contrastive loss, which could be approximately computed in linear time and space complexity with respect to the graph size. We provide theoretical analysis to justify the effectiveness and rationality of the proposed methods. Experiments on various datasets with different scales and properties demonstrate that in spite of its simplicity, LOCAL-GCL achieves quite competitive performance in self-supervised node representation learning tasks on graphs with various scales and properties.

## 1 INTRODUCTION

Self-supervised learning has achieved remarkable success in learning informative representations without using costly handcrafted labels (van den Oord et al., 2018; Devlin et al., 2019; Banville et al., 2021; He et al., 2020; Chen et al., 2020; Grill et al., 2020; Zhang et al., 2021; Gao et al., 2021). Among current self-supervised learning paradigms, InfoNCE loss (van den Oord et al., 2018) based multi-view contrastive methods (He et al., 2020; Chen et al., 2020; Gao et al., 2021) are recognized as the most widely adopted ones, due to their solid theoretical foundations and strong empirical results. Generally, contrastive learning aims at maximizing the agreement between the latent representations of two views (e.g. through data augmentation) from the same input, which essentially maximizes the mutual information between the two representations (Poole et al., 2019). Inheriting the spirits of contrastive learning on vision tasks, similar methods have been developed to deal with graphs and bring up promising results on common node-level classification benchmarks (Velickovic et al., 2019; Hassani & Ahmadi, 2020; Zhu et al., 2020b; 2021).

The challenge, however, is that prevailing contrastive learning methods rely on predefined augmentation techniques for generating positive pairs as informative training supervision. Unlike grid-structured data (e.g., images or sequences), it is non-trivial to define well-posed augmentation approaches for graph-structured data Zhu et al. (2021); Zhang et al. (2021). The common practice adopted by current methods resorts to random perturbation on input node features and graph structures (You et al., 2020), which might unexpectedly violate the underlying data generation and change the semantic information (Lee et al., 2021). Such an issue plays as a bottleneck limiting the practical efficacy of contrastive methods on graphs. Apart from this, the InfoNCE loss function computes all-pair distance for in-batch nodes as negative pairs for contrasting signals (Zhu et al., 2020b; 2021), which induces quadratic memory and time complexity with respect to the batch size. Given that the model is preferred to be trained in a full-graph manner (i.e., batch size = graph size) since the graph structure information might be partially lost through mini-batch partition, such a nature heavily constrains contrastive methods for scaling to large graphs.

Some recent works seek negative-sample-free methods to resolve the scalability issue by harnessing asymmetric structures (Thakoor et al., 2021) or feature-level decorrelation objectives (Zhang et al., 2021). However, these methods either lack enough theoretical justification (Thakoor et al., 2021) or

necessitate strong assumptions on the data distributions (Zhang et al., 2021). Moreover, they still require data augmentations to generate two views of the input graph, albeit non-contrastive and free from negative sampling. Some other works construct positive examples using the target's $k$-nearest-neighbors (kNN) in the latent space (Dwibedi et al., 2021; Koohpayegani et al., 2021; Lee et al., 2021). Nonetheless, the computation of nearest neighbors could be cumbersome, time-consuming, and, therefore hard to scale.

**Presented Work.** To cope with the dilemmas above, in this paper we introduce **Local**ized **G**raph **C**ontrastive **L**earning (LOCAL-GCL in abbreviation), a light and **augmentation-free** contrastive model for self-supervised node-level representation learning on graphs. LOCAL-GCL benefits from two key designs. **First**, it does not rely on data augmentation to construct positive pairs. Instead, inspired by the graph homophiliy theory (McPherson et al., 2001), it directly treats the first-order neighboring nodes as the positive examples of the target node. This not only increases the number of positive examples for each node but also helps our model get rid of complicated data augmentations. Besides, the computation of positive pairs can be performed in linear time and space complexity w.r.t the graph size, bringing no additional cost for the model. **Second**, to deal with the quadratic complexity curse of contrastive loss (i.e., InfoNCE loss (van den Oord et al., 2018)), we propose a surrogate loss function in place of the negative loss term in the vanilla InfoNCE loss, which could be efficiently and accurately approximated within linear time and space complexity (Rahimi et al., 2007; Liu et al., 2020; Yu et al., 2016). Such a design greatly improves the efficiency of our model.

We evaluate the proposed methods on seven public node classification benchmarks with various scales. The empirical results demonstrate that though not using any graph augmentations, our method achieves state-of-the-art performance on six of seven datasets. On the challenging `Ogbn-Arxiv` dataset, our method can also give a competitive performance with a much training speed compared with other scalable models. Experiments on three heterophily graphs demonstrate that besides homophily graphs, LOCAL-GCL can also perform well on graphs with low homophily ratios.

**We summarize the highlights of this paper as follows:**

**1)** We introduce LOCAL-GCL, a simple model for contrastive learning on graphs, where the positive example is fabricated using the first-order neighborhood of each node. This successfully frees node-level contrastive learning methods from unjustified graph augmentations.

**2)** To overcome the quadratic complexity curse of contrastive learning, we propose a kernelized contrastive loss computation that can precisely approximate the original loss function within linear complexity w.r.t. graph size. This significantly reduces the training time and memory cost of contrastive learning on large graphs.

**3)** Experimental results show that without data augmentation and other cumbersome designs LOCAL-GCL achieves quite competitive results on a variety of graphs of different scales and properties. Furthermore, LOCAL-GCL demonstrates a better balance of model performance and efficiency than other self-supervised methods.

## 2 BACKGROUND AND RELATED WORKS

### 2.1 CONTRASTIVE REPRESENTATION LEARNING

Inspired by the great success of contrastive methods in learning image representations (van den Oord et al., 2018; Hjelm et al., 2019; Tian et al., 2020; He et al., 2020; Chen et al., 2020), recent endeavors develop similar strategies for node-level tasks in graph domain (Velickovic et al., 2019; Hassani & Ahmadi, 2020; Zhu et al., 2020b; 2021). Among graph contrastive learning methods, the most popular methods should be those based on the InfoNCE loss (van den Oord et al., 2018) due to their simple concepts and better empirical performance. InfoNCE-based graph contrastive learning methods, including GRACE (Zhu et al., 2020b) and GCA (Zhu et al., 2021) aim to maximize the similarity of positive node-node (or graph-graph) pairs (e.g., two views generated via data augmentation) and minimize the similarity of negative ones (e.g., other nodes/graphs within the current batch). However, they require well-designated data augmentations that could positively inform downstream tasks (Tsai et al., 2021). The quadratic complexity also limits their applications to larger batch sizes/datasets.

## 2.2 AUGMENTATION-FREE SELF-SUPERVISED LEARNING ON GRAPHS

Besides graph contrastive learning (Zhu et al., 2020b; 2021) and the similar two-branched models (Thakoor et al., 2021; Zhang et al., 2021) that use graph augmentations to create positive pairs, there is another line of work exploring the rich structure information in the graph to create self-supervised signals. Graph AutoEncoders (Kipf & Welling, 2016), for example, learn node embeddings unsupervisedly through reconstructing the adjacency matrix, while GMI (Peng et al., 2020) maximizes the mutual information of both features and edges between inputs and outputs. Inspired by the success of contrastive learning, some recent work explores constructing positive pairs using reliable graph information instead of data augmentations. For example, SUGRL (Mo et al., 2022) employs two encoder models: one is a GCN, and the another is an MLP, to generate two sets of node embedding from different sources. Then for each node, the positive pair can be constructed using its GCN output and MLP output. AFGRL (Lee et al., 2021) and AF-GCL (Wang et al., 2022) treat nodes in the target node's multi-hop neighborhood as candidate positive examples and use well-designed similarity measures to select the most similar nodes as positive examples. LOCAL-GCL, by contrast, treats all neighbors equally without discrimination. This not only simplifies the model design but also offers a theoretical guarantee for the effectiveness of the proposed model. Empirical results also demonstrate that our method can achieve better performance than them.

## 2.3 RANDOM FOURIER FEATURES

Though InfoNCE-loss-based based methods are successful in self-supervised node representation learning, their quadratic time and memory complexity with respect to the graph size prevent them from being applied to graphs with tens of thousands of nodes. This paper seeks to address this issue by optimizing a surrogate loss function with Random Fourier Features (RFF or Random Features in short) (Rahimi et al., 2007; Liu et al., 2020). RFF is an effective technique for enhancing the scalability of kernel methods such as SVM, ridge regression (Avron et al., 2017) and independence test (Zhang et al., 2018; Li et al., 2021). Also, recently, RFF has been adopted to develop linear Transformers by approximating the softmax attention (Choromanski et al., 2020; Peng et al., 2021). Given $d$-dimensional vectors $\boldsymbol{x}$ and $\boldsymbol{y}$ and a shift-invariant kernel $\kappa(\cdot)$, RFF constructs an explicit mapping $\psi$: $\mathbb{R}^d \to \mathbb{R}^D$, such that $\kappa(\boldsymbol{x}, \boldsymbol{y}) \approx \psi(\boldsymbol{x})^\top \psi(\boldsymbol{y})$, which reduces the quadratic computation cost of the kernel matrix to a linear one w.r.t data size. Generally, given a positive definite shift-invariant kernel $\kappa(\boldsymbol{x}, \boldsymbol{y}) = f(\boldsymbol{x} - \boldsymbol{y})$, the Fourier transform $p$ of kernel $\kappa$ is $p(\boldsymbol{\omega}) = \frac{1}{2\pi} \int e^{-j\boldsymbol{\omega}'\Delta} k(\Delta) \mathrm{d}\Delta$, where $\Delta = \boldsymbol{x} - \boldsymbol{y}$. Then we could draw $D$ i.i.d. samples $\boldsymbol{\omega}_1, \cdots, \boldsymbol{\omega}_D \in \mathbb{R}^d$ from $p$, and $\psi(\boldsymbol{x})$ is given by:

$$\psi(\boldsymbol{x}) = \frac{\left[\cos(\boldsymbol{\omega}_1^\top \boldsymbol{x}), \cdots, \cos(\boldsymbol{\omega}_D^\top \boldsymbol{x}), \sin(\boldsymbol{\omega}_1^\top \boldsymbol{x}), \cdots, \sin(\boldsymbol{\omega}_D^\top \boldsymbol{x})\right]^\top}{\sqrt{D}}. \tag{1}$$

Let $\boldsymbol{W} = [\boldsymbol{\omega}_1, \cdots, \boldsymbol{\omega}_D] \in \mathbb{R}^{d \times D}$ be a linear transformation matrix, one may realize that the computation of Eq. 1 entails computing $\boldsymbol{W}^\top \boldsymbol{x}$. Specifically, when $\kappa(\cdot)$ is a standard Gaussian kernel (a.k.a., RBF kernel), each entry of $\boldsymbol{W}$ can be directly sampled from a standard Gaussian distribution. The improved variants of RFF mainly concentrate on different ways to build the transformation matrix $\boldsymbol{W}$, so as to further reduce the computational cost (Le et al., 2013) or lower the approximation variance (Yu et al., 2016).

## 3 CONTRASTIVE LEARNING ON GRAPHS WITH INFONCE LOSS

The InfoNCE (van den Oord et al., 2018) loss has been employed in various areas and has shown great power in learning informative representations in a self-supervised manner. Given the embedding of a target data point $\boldsymbol{z}_i$, together with one positive embedding $\boldsymbol{z}_i^+$ and a set of negative embeddings $\{\boldsymbol{z}_j^-\}_{j=1}^{M-1}$, the InfoNCE loss aims at discriminating the positive pair $(\boldsymbol{z}_i, \boldsymbol{z}_i^+)$ from the negative pairs $\{(\boldsymbol{z}_i, \boldsymbol{z}_j^-)\}_{j=1}^{M-1}$ via the following contrastive objective function:

$$\mathcal{L}_{\text{InfoNCE}}(i) = -\log \frac{\exp(f(\boldsymbol{z}_i, \boldsymbol{z}_i^+)/\tau)}{\exp(f(\boldsymbol{z}_i, \boldsymbol{z}_i^+)/\tau) + \sum_{j=1}^{M-1} \exp(f(\boldsymbol{z}_i^+, \boldsymbol{z}_j^-)/\tau)}, \tag{2}$$

where $f(\cdot, \cdot)$ is a similarity measure, usually implemented as the simple dot product $f(\boldsymbol{x}, \boldsymbol{y}) = \boldsymbol{x}^\top \boldsymbol{y}$ (He et al., 2020; Chen et al., 2020), and $\tau$ is the temperature hyperparameter. Note that $\boldsymbol{z}_i, \boldsymbol{z}_i^+$

and $z_i^-$ are all $\ell_2$ normalized to have a **unit norm**, i.e., $\|z_i\|_2^2 = 1$. In the context of node-level contrastive learning, each $z_i$ denotes the embedding of node $i$, which is the output of an encoder model which takes the node feature $x_i$ and the graph structure as input. As demonstrated in Eq. 2, the InfoNCE loss is composed of two terms: 1) the numerator term on the positive pair that maximizes the similarity between positive pairs (positive term); 2) the denominator term on $M - 1$ negative pairs and one positive pair (for simplicity, we call all the $M$ terms in the denominator "negative terms") that encourages the embeddings of negative pairs to be distinguished (negative term). Like the study of contrastive learning on vision and language, recent node-level contrastive learning research mainly focuses on how to construct/select more informative positive and negative examples.

The most popular method for constructing **positive examples** on graphs is through data augmentation. For example, MVGRL (Hassani & Ahmadi, 2020) uses graph diffusion to generate a fixed, augmented view of the studied graph. GCC (Qiu et al., 2020) resorts to random walk to sample subgraphs of the anchored node as positive examples. GRACE (Zhu et al., 2020b) and GCA (Zhu et al., 2021) employ random graph augmentation–*feature masking* and *edge removing* to generate two views of the input graph before every training step. However, recent studies have shown that existing graph augmentations might unexpectedly change or lose the semantic information (Lee et al., 2021; Wang et al., 2022) of the target node, hurting the learned representation. This motivates us to investigate the possibility of constructing positive examples without using random, ambiguous graph augmentations.

Compared with the construction of positive examples, **negative examples** are much easier to select. A common adopted and effective practice is to use all other nodes in the graph as negative examples (Zhu et al., 2020b; 2021; Zhang et al., 2021). However, the computation of the negative term of InfoNCE has $\mathcal{O}(|\mathcal{V}|^2 d)$ time complexity and $\mathcal{O}(|\mathcal{V}|^2)$ memory complexity, where $|\mathcal{V}|$ is the number of nodes and $d$ is the embedding dimension. Note that the real-world graphs are usually very large, Eq. 2 can hardly be used in a full-graph training manner. One plausible remedy is to use down-sampling to zoom in on a fraction of negative samples for once feedforward computation. However, theoretical analysis demonstrates InfoNCE loss benefits from a large number of negative examples (Poole et al., 2019), and empirical observations show that reducing the number of negative examples leads to worse performance on downstream tasks (Thakoor et al., 2021) (also see our experiments in Sec. 5.4). As a result, it would be promising to devise an efficient and scalable computation method for the contrastive loss above, without reducing the number of negative examples.

In the next section, we propose Localized Graph Contrastive Learning with Kernel Approximation (LOCAL-GCL in short) to mitigate the above issues. LOCAL-GCL distinguishes itself from previous contrastive methods in the following two aspects: 1) instead of constructing positive examples for target nodes using data augmentations or other techniques, we directly treat all the first-order neighbors of each target node as its positive examples; 2) we introduce a Gaussian kernelized surrogate loss function, which could be accurately and efficiently approximated with linear time and memory complexity in place of the vanilla InfoNCE loss.

## 4 METHODOLOGY

### 4.1 ANCHORING NODE CONTRASTIVE LEARNING ON TOPOLOGY

Contrastive learning is firstly investigated in unsupervised visual representation learning where the data points are i.i.d. sampled from the data distribution, and the ground-truth labels are unknown during training. As a result, we have to employ well-defined augmentations to create two views of the same image that are likely to share the same ground-truth label. Different from images or texts where each input data is often assumed i.i.d. generated from a certain distribution and there is no explicit prior knowledge about the relationship among different instances, in the graph domain, nodes within the same graph are highly inter-dependent given the input graph structure. That is, the graph structure could provide additional information that reflects the affinity of node pairs. Furthermore, in network (graph) science, it is widely acknowledged that many real-world graphs (such as social networks (Wu et al., 2019; McPherson et al., 2001) and citation networks (Ciotti et al., 2016)) conform with the homophily phenomenon that similar nodes may be more likely to attach than dissimilar ones (Zhu et al., 2003; McPherson et al., 2001). Such a phenomenon also inspires the core principle followed by the designs of modern graph neural networks, i.e., connected nodes should have similar embeddings in the latent space (Zhu et al., 2020a). This motivates us to directly treats all the first-order neighboring

nodes of the target node as positive examples. Formally, the loss for the positive pairs of node $i$ can be written as

$$\mathcal{L}_{\text{pos}}(i) = -\log \sum_{j \in \mathcal{N}(i)} \exp(\boldsymbol{z}_i^\top \boldsymbol{z}_j / \tau) / |\mathcal{N}(i)|. \tag{3}$$

For the negative term, we can directly use all nodes in the graph as negative examples. Then the loss for the negative pairs of node $i$ can be formulated as:

$$\mathcal{L}_{\text{neg}}(i) = \log \sum_{k \in \mathcal{V}} \exp(\boldsymbol{z}_i^\top \boldsymbol{z}_k / \tau). \tag{4}$$

The overall loss function is simply the sum of $\mathcal{L}_{\text{pos}}$ and $\mathcal{L}_{\text{neg}}$, then averaged over all nodes:

$$\mathcal{L}_{\text{Local-GCL}} = -\frac{1}{|\mathcal{V}|} \sum_{i=1}^{N} \log \frac{\sum_{j \in \mathcal{N}(i)} \exp(\boldsymbol{z}_i^\top \boldsymbol{z}_j / \tau) / |\mathcal{N}(i)|}{\sum_{k \in \mathcal{V}} \exp(\boldsymbol{z}_i^\top \boldsymbol{z}_k / \tau)}. \tag{5}$$

Then we'd like to provide a theoretical analysis of the effectiveness of constructing positive examples directly with the first-order neighbors. First, let's introduce some notations that will be used in the following analysis. We denote a graph by $\mathcal{G} = (\mathcal{V}, \mathcal{E})$, where $\mathcal{V}$ is the node st and $\mathcal{E}$ is the edge set. The adjacency matrix and degree matrix are denoted by $\boldsymbol{A}$ and $\boldsymbol{D}$, respectively. Let $\tilde{\boldsymbol{A}} = \boldsymbol{D}^{-1/2} \boldsymbol{A} \boldsymbol{D}^{-1/2}$ be the symmetric normalized adjacency matrix, and $\boldsymbol{L} = \boldsymbol{I} - \tilde{\boldsymbol{A}}$ be the symmetric normalized graph Laplacian matrix. Denote the eigenvalues of $\boldsymbol{L}$ in Zn ascending order by $\{\lambda_i\}_{i=1}^{|\mathcal{V}|}$. Finally we denote the node embedding matrix $\boldsymbol{Z} = \{\boldsymbol{z}_i\}_{i=1}^{|\mathcal{V}|} \in \mathbb{R}^{|\mathcal{V}| \times d}$ and the one-hot label matrix by $\boldsymbol{Y} = \{\boldsymbol{y}_i\}_{i=1}^{|\mathcal{V}|} \in \mathbb{R}^{|\mathcal{V}| \times c}$. Without loss of generality we assume $d \leq |\mathcal{V}|$.

We then give a formal definition of graph homophily ratio $\phi$:

**Definition 1.** *(Graph Homophily Ratio) For a graph $\mathcal{G} = (\mathcal{V}, \mathcal{E})$ with adjacency matrix $\boldsymbol{A}$, its homophily ratio $\phi$ is defined as the probability that two connected nodes share the same label:*

$$\phi = \frac{\sum_{i,j \in \mathcal{V}} A_{ij} \cdot \mathbb{1}[\boldsymbol{y}_i = \boldsymbol{y}_j]}{\sum_{i,j \in \mathcal{V}} A_{ij}} = \frac{\sum_{i,j \in \mathcal{V}} A_{ij} \cdot \mathbb{1}[\boldsymbol{y}_i = \boldsymbol{y}_j]}{|\mathcal{E}|} \tag{6}$$

With the conclusions in Balestriero & LeCun (2022) and HaoChen et al. (2021), which build connections between contrastive loss and spectral method, the following theorem guarantees the linear classification error of the embeddings learned from LOCAL-GCL:

**Theorem 1.** *Let $\boldsymbol{Z}^*$ be the global minimizer of Eq. 5, then for any labeling function $\hat{y} : \mathbb{V} \to \mathbb{R}^c$ with graph homophily $\phi$, there exists a linear classifier $\boldsymbol{B}^* \in \mathbb{R}^{d \times c}$ with norm $\|\boldsymbol{B}^*\|_F \leq 1/(1 - \lambda_d)$ such that*

$$\mathbb{E}_{i \in \mathcal{V}} \left[ \|\hat{y}(i) - \boldsymbol{B}^{*\top} \boldsymbol{z}_i^* \|_2^2 \right] \leq \frac{1 - \phi}{\lambda_{d+1}} \tag{7}$$

See proof in Appendix A.1. Theorem 1 demonstrates that the linear classification accuracy of the learned embeddings through LOCAL-GCL is bounded by the homophily ratio $\phi$ and the $d+1$ smallest eigenvalue $\lambda_{d+1}$. Specifically, the larger the homophily ratio, the smaller the prediction error. Besides, Eq. 7 indicates that a larger embedding dimension can lead to better classification accuracy, which is also validated empirically in Sec. 5.

## 4.2 FAST AND EFFICIENT APPROXIMATION FOR THE NEGATIVE LOSS

We next probe into how to reduce the computation complexity of the negative loss term $\mathcal{L}_{\text{neg}} = \sum_{i \in \mathcal{V}} \mathcal{L}_{\text{neg}}(i)$, for which the $|\mathcal{V}|^2$ pair-wise similarities can be efficiently and precisely approximated with efficient computation methods for kernel functions (Rahimi et al., 2007; Liu et al., 2020).

We notice that the pair-wise similarity (in dot-product format) in Eq. 4 is essentially a Gaussian kernel function, i.e.,

$$\exp(\boldsymbol{z}_i^\top \boldsymbol{z}_j / \tau) = \exp(\frac{2 - \|\boldsymbol{z}_i - \boldsymbol{z}_j\|_2^2}{2\tau}) = e * \exp(\frac{-\|\boldsymbol{z}_i - \boldsymbol{z}_j\|}{2\tau}) = e \cdot \kappa^G(\boldsymbol{z}_i, \boldsymbol{z}_k; \sqrt{\tau}), \tag{8}$$

where $\kappa^G(\boldsymbol{z}_i, \boldsymbol{z}_k; \sqrt{\tau})$ is the Gaussian kernel function with bandwidth $\sqrt{\tau}$. This motivates us to seek for a projection function $\psi(\boldsymbol{x})(\mathbb{R}^d \to \mathbb{R}^{2D})$ such that $\kappa^G(\boldsymbol{x}, \boldsymbol{y}; \sqrt{\tau}) \approx \psi(\boldsymbol{x})^\top \psi(\boldsymbol{y})$. Then we are able to use a surrogate negative loss function as a remedy of the previous one:

$$\mathcal{L}_{neg} \triangleq \frac{1}{|\mathcal{V}|} \sum_{i \in \mathcal{V}} \left( \log \sum_{j \in \mathcal{V}} \psi(\boldsymbol{h}_i)^\top \psi(\boldsymbol{h}_j) \right) = \frac{1}{|\mathcal{V}|} \sum_{i \in \mathcal{V}} \left( \log \left( \psi(\boldsymbol{h}_i)^\top \sum_{j \in \mathcal{V}} \psi(\boldsymbol{h}_j) \right) \right). \quad (9)$$

Once we obtain all the projected vectors $\{\psi(\boldsymbol{h}_i)\}_{i=1}^{|\mathcal{V}|}$, the summation term $\sum_{j \in \mathcal{V}} \psi(\boldsymbol{h}_j)$ in Eq. 9 could be calculated within $\mathcal{O}(|\mathcal{V}|D)$ in advance (where $D$ is the dimension of projected vectors). In addition to the $\mathcal{O}(|\mathcal{V}|D)$ cost for computing the loss, the overall computational cost will be $\mathcal{O}(|\mathcal{V}|D + |\mathcal{V}|D) = \mathcal{O}(VD)$, which is linear to the number of nodes. Then we discuss how to formulate the projection function $\psi(\cdot)$ below.

**Linear-Order Projection**    The theory of Random Fourier Features (Rahimi et al., 2007) introduced in Sec. 2.3 demonstrates that a Gaussian kernel function $\kappa^G(\boldsymbol{x}, \boldsymbol{y}; \sqrt{\tau})$ could be unbiasedly estimated with $\psi(\boldsymbol{x})^\top \psi(\boldsymbol{y})$, and the projection function $\psi(\boldsymbol{x})$ is defined as follows:

$$\psi(\boldsymbol{x}) = \frac{[\cos(\boldsymbol{W}^\top \boldsymbol{x}), \sin(\boldsymbol{W}^\top \boldsymbol{x})]}{\sqrt{D}} = \frac{\left[ \cos(\boldsymbol{\omega}_1^\top \boldsymbol{x}), \cdots, \cos(\boldsymbol{\omega}_D^\top \boldsymbol{x}), \sin(\boldsymbol{\omega}_1^\top \boldsymbol{x}), \cdots, \sin(\boldsymbol{\omega}_D^\top \boldsymbol{x}) \right]^\top}{\sqrt{D}}.$$
$$(10)$$

where $\boldsymbol{W} = [\boldsymbol{\omega}_1^\top, \cdots, \boldsymbol{\omega}_D^\top]$, and each $\boldsymbol{\omega}_i$ is sampled from the Gaussian distribution $p(\boldsymbol{\omega}) = \mathcal{N}(\boldsymbol{0}, \tau\boldsymbol{I})$. $D$ is the number of total samples. Usually, the larger the sampling number $D$ is, the more accurate the approximation will be:

**Theorem 2.** *Let $\{\boldsymbol{\omega}_i\}_{i=1}^D$ be i.i.d samples from Gaussian distribution $\mathcal{N}(\boldsymbol{0}, \tau\boldsymbol{I})$, and $\psi(\boldsymbol{x})$ is given by Eq. 10, then with probability at least $1-\varepsilon$, the approximation error $\Delta = |\psi(\boldsymbol{h}_i)^\top \psi(\boldsymbol{h}_j) - \kappa^G(\boldsymbol{h}_i, \boldsymbol{h}_j)|$ will be bounded by $\mathcal{O}\left( \frac{1-\exp(-4/\tau)}{\sqrt{2D\varepsilon}} \right)$.*

Theorem 2 suggests that the Gaussian kernel function could be accurately approximated with Random Fourier Features as long as we sample enough number of linear transformation vectors $\boldsymbol{\omega}$ (i.e., $D$ should be large enough). Note that the computation of linear projection $\boldsymbol{W}\boldsymbol{h}_i$ ($i = 1 \in \mathcal{V}$) requires additional $\mathcal{O}(|\mathcal{V}|dD)$ time. A large number of the projection dimension $D$ makes the computation of Eq. 9 still expensive for high-dimensional data.

**Log-Order Projection**    To handle the above issues, we resort to Structured Orthogonal Random Features (SORF) (Yu et al., 2016), another Random Feature technique that imposes structural and orthogonality on the linear transformation matrix $\boldsymbol{W}$. Different from the vanilla RFF which directly samples linear transformation vectors $\boldsymbol{\omega}$ form normal distribution to construct the transformation matrix $\boldsymbol{W}_{\text{rff}} = [\boldsymbol{\omega}_1^\top, \cdots, \boldsymbol{\omega}_D^\top] \in \mathbb{R}^{d \times D}$, SORF assumes $D = d$ and constructs a structured orthogonal transformation matrix through the continued product of a series of structured matrixes $\boldsymbol{W}_{\text{sorf}} = \frac{\sqrt{d}}{\sigma} \boldsymbol{H}\boldsymbol{D}_1\boldsymbol{H}\boldsymbol{D}_2\boldsymbol{H}\boldsymbol{D}_3$, where $\boldsymbol{D}_i \in \mathbb{R}^{d \times d}, i = 1, 2, 3$, are diagonal "sign-flipping" matrices, with each diagonal entry sampled from the Rademacher distribution, and $\boldsymbol{H} \in \mathbb{R}^{d \times d}$ is the normalized Walsh-Hadamard matrix. By this definition, the projected dimension is restricted to $d$, but could be extended to any dimension by concatenating multiple independently generated features or simply using a proportion of them. The stand-out merit of SORF is that it can be computed in $\mathcal{O}(|\mathcal{V}|D \log d)$ time using fast Hadamard transformation (Fino & Algazi, 1976), and hardly requires extra memory cost using in-place operations (Yu et al., 2016). This further reduces its complexity and endows our method with desirable scalability to not only larger dataset sizes but also larger embedding dimensions. If not specified, in the following section, the term LOCAL-GCL denotes our method equipped with SORF to approximate the negative loss.

## 5    EXPERIMENTS

We conduct experiments to evaluate the proposed method by answering the following questions:

- **RQ1**: How does LOCAL-GCL perform compared with other self-supervised learning methods on graphs with different properties?

- **RQ2**: How do the specific designs of LOCAL-GCL, such as the embedding dimension and projection dimension, affect its performance?

- **RQ3**: What's the empirical memory and time consumption of LOCAL-GCL compared with prior works? Is LOCAL-GCL able to scale to real-world large-scale graphs with satisfying performance?

## 5.1 EXPERIMENTAL SETUPS

**Datasets.** We evaluate LOCAL-GCL on various datasets with different scales and properties. Following prior works (Zhu et al., 2020b; Zhang et al., 2021) on self-supervised node representation learning, we adopt the 7 common small-scale benchmarking graphs: `Cora`, `Citeseer`, `Pubmed`, `Amazon-Computer`, `Amazon-Photo`, `Coauthor-CS`, and `Coauthor-Physics`. To evaluate the performance and scalability of LOCAL-GCL on larger graphs, we also adopt `Ogbn-Arxiv` with about 170k nodes, on which a lot of methods fail to scale due to memory issues. Furthermore, we adopt three widely used heterophily graphs, `Chameleon`, `Squirrel`, and `Actor` to evaluate the generalization ability of LOCAL-GCL on graphs where the graph homophily assumption does not hold. The detailed introduction and statistics of these datasets are presented in Appendix C.1

**Baselines.** We consider representative prior self-supervised models for comparison. We classify previous methods into two types: 1) **Augmentation-based**, which uses data augmentations to generate positive or negative pairs. 2) **Augmentation-free**, which uses other information rather than any form of data augmentation to create self-supervised signals. For augmentation-based baselines, we consider DGI (Velickovic et al., 2019), MVGRL (Hassani & Ahmadi, 2020), GRACE (Zhu et al., 2020b), GCA Zhu et al. (2021), BGRL (Thakoor et al., 2021) and CCA-SSG (Zhang et al., 2021). For augmentation-free baselines, we consider GMI (Peng et al., 2020), SUGRL (Mo et al., 2022), AFGRL (Lee et al., 2021), AF-GCL (Wang et al., 2022).

**Evaluation Protocols.** We follow the linear evaluation scheme in previous works (Velickovic et al., 2019; Hassani & Ahmadi, 2020; Zhu et al., 2021; Zhang et al., 2021): For each dataset, i) we first train the model on all the nodes in a graph without supervision by optimizing the objective in Eq. 5; ii) after the training ends, we freeze the parameters of the encoder and obtain all the nodes' embeddings, which are subsequently fed into a linear classifier (i.e., a logistic regression model) to generate a predicted label for each node. In the second stage, only nodes in the training set are used for training the classifier, and we report the classification accuracy on testing nodes.

**Implementation Details.** The model is implemented with PyTorch and DGL (Wang et al., 2019). All experiments are conducted on an NVIDIA V100 GPU with 16 GB memory unless specified. We adopt `structure-net`[1] to do fast Walsh-Hadamard transformation, which enables much faster forward and backward computation with CUDA accelerations. We use the Adam optimizer (Kingma & Ba, 2015) for both self-supervised pretraining training and linear evaluation using logistic regression. Following previous works (Zhu et al., 2021; Zhang et al., 2021; Thakoor et al., 2021; Wang et al., 2022), we the random 1:1:8 split for `Amazon-Computer`, `Amazon-Photo`, `Coauthor-CS` and `Amazon-Computer`, and use the public recommended split for the remaining datasets. For each experiment, we report the average test accuracy with the standard deviation over 20 random initialization. If not specified, we use a two-layer GCN model as the encoder to generate node embeddings. More detailed hyperparameter settings for each dataset can be found in Appendix C.2

## 5.2 NUMERICAL RESULTS

**Results on common graphs.** We first report the results of node classification tasks on small-scale citation networks and social networks in Table 1. We see that although not relying on data augmentations or other complicated operations to create self-supervised signal, LOCAL-GCL performs competitively with our self-supervised baselines, achieving state-of-the-art performance in 6 out of 7 datasets. It is worth noting that the competitive InfoNCE-loss based contrastive methods GRACE and GCA suffer from OOM on `Coauthor-Physics` datasets due to the $\mathcal{O}(|\mathcal{V}|^2 d)$ space complexity, while LOCAL-GCL can avoid such an issue thanks to the linear approximation of the negative loss.

---

[1]https://github.com/HazyResearch/structured-nets

Table 1: Comparison of self-supervised methods on benchmarking graphs. We group each method according to whether it relies on graph augmentation.

| | Methods | Cora | Citeseer | Pubmed | Computer | Photo | CS | Physics |
|---|---|---|---|---|---|---|---|---|
| Aug-based | DGI | 82.3±0.6 | 71.8±0.7 | 76.8±0.6 | 83.95±0.47 | 91.61±0.22 | 92.15±0.63 | 94.51±0.52 |
| | MVGRL | 83.5±0.4 | 73.3±0.5 | 80.1±0.7 | 87.52±0.11 | 91.74±0.07 | 92.11±0.12 | 95.33±0.03 |
| | GRACE | 81.9±0.4 | 71.2±0.5 | 80.6±0.4 | 86.25±0.25 | 92.15±0.24 | 92.93±0.01 | OOM* |
| | GCA | 82.3±0.4 | 72.1±0.4 | 80.7±0.5 | 87.85±0.31 | 92.49±0.09 | 93.10±0.01 | OOM |
| | BGRL | 82.7±0.6 | 71.1±0.8 | 79.6±0.5 | 89.69±0.37 | 93.07±0.28 | 92.59±0.17 | 95.48±0.08 |
| | CCA-SSG | 84.2±0.4 | 73.1±0.3 | 81.6±0.4 | 88.74±0.28 | 93.14±0.14 | 93.31±0.22 | 95.38±0.06 |
| Aug-free | GMI | 82.4±0.6 | 71.7±0.2 | 79.3±1.0 | 84.22±0.52 | 90.73±0.24 | OOM | OOM |
| | SUGRL | 83.4±0.5 | 73.0±0.4 | 81.9±0.3 | 88.93±0.21 | 93.07±0.15 | 92.83±0.23 | 95.38±0.11 |
| | AFGRL | 81.3±0.2 | 68.7±0.3 | 80.6±0.4 | **89.88±0.33** | **93.22±0.28** | 93.27±0.17 | OOM |
| | AF-GCL | 83.2±0.2 | 72.0±0.4 | 79.1±0.8 | 89.68±0.19 | 92.49±0.31 | 91.92±0.10 | 95.12±0.15 |
| | LOCAL-GCL | **84.5±0.4** | **73.6±0.4** | **82.1±0.5** | 88.81±0.37 | **93.25±0.40** | **94.90±0.19** | **96.33±0.13** |

* OOM indicates out-of-memory on an NVIDIA-V100 GPU of 16G memory.

Table 2: Performance on `Ogbn-Arxiv` dataset. As recommended, we report both the validation accuracy and test accuracy.

| Methods | Validation | Test |
|---|---|---|
| DGI | 71.21±0.23 | 70.32±0.25 |
| GMI | OOM | OOM |
| MVGRL | OOM | OOM |
| GRACE | OOM | OOM |
| GCA | OOM | OOM |
| BGRL | **72.71±0.22** | **71.54±0.17** |
| CCA-SSG | 72.31±0.18 | 71.21±0.20 |
| AFGRL | OOM | OOM |
| LOCAL-GCL | 72.29±0.25 | 71.34±0.25 |

Table 3: Performance on Heterophily graphs. Results of baseline methods are taken from Wang et al. (2022)

| Methods | Chameleon | Squirrel | Actor |
|---|---|---|---|
| DGI | 60.27±0.70 | 42.22±0.63 | 28.30±0.76 |
| GMI | 52.81±0.63 | 35.25±1.21 | 27.28±0.87 |
| MVGRL | 53.81±1.09 | 38.75±1.32 | 32.09±1.07 |
| GRACE | 61.24±0.53 | 41.09±0.85 | 28.27±0.43 |
| GCA | 60.94±0.81 | 41.53±1.09 | 28.89±0.50 |
| BGRL | 64.86±0.63 | 46.24±0.70 | 28.80±0.54 |
| AF-GCL | 65.28±0.53 | 52.10±0.67 | 28.94±0.69 |
| LOCAL-GCL | **68.74±0.49** | **52.94±0.88** | **33.91±0.57** |

**Results on Ogbn-Arxiv.** Then, we evaluate the effectiveness and scalability of LOCAL-GCL on large-scale graphs taking `Ogbn-Arxiv` as an example. Following the practice in Thakoor et al. (2021), we expand the encoder to a 3-layer GCN model. We report the validation and test accuracy of baseline models and ours in Table 2. As demonstrated, many baseline methods cannot run in a full graph manner (on a GPU with 16GB memory). Compared with other scalable methods, LOCAL-GCL can give a competitive performance on `Ogbn-Arxiv`.

**Results on Heterophily graphs.** Finally, we investigate the performance on non-homophily graphs, a much more challenging task as directly using first-order neighbors as positive examples without discriminating might be harmful to non-homophily graphs. The results on the three heterophily graphs are presented in Table 3. Counterintuitively, LOCAL-GCL achieves amazing performance, outperforming all the baseline methods. This is probably due to the following reasons: 1) Even though connected nodes may not share the same label, as long as the neighborhood distributions for different classes are different, LOCAL-GCL is able to recognize the neighborhood patterns of different classes and make the node embeddings for different classes distinguishable. This is also justified in one recent work showing that a GCN model can still perform well on heterophily graphs (Ma et al., 2022). 2) Data augmentation-based methods tend to keep the low-frequency information while discarding the high-frequency one (Wang et al., 2022; Liu et al., 2022), while high-frequency information is much more important for classification on heterophilic graphs (Bo et al., 2021). By contrast, edge-wise positive pairs enable LOCAL-GCL to learn the differences between connected nodes better, thus benefiting heterophilic graphs. We provide more explanations for this point in Appendix C.4.

### 5.3 ABLATIONS

As demonstrated in Theorem 1 and Sec. 4.2, the embedding dimension $d$ and projection dimension $D$ should be two critical hyperparameters affecting the model's performance. In this section, we test

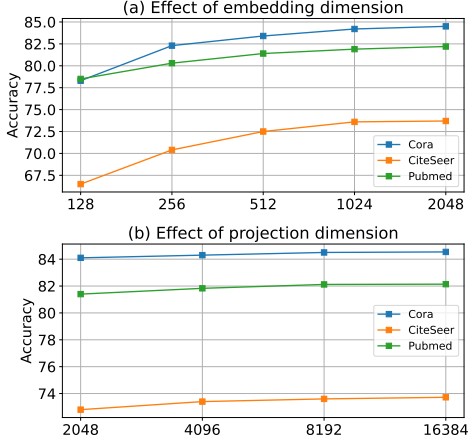

Figure 1: Sensitivity analysis of embedding dimension $d$ and projection dimension $D$.

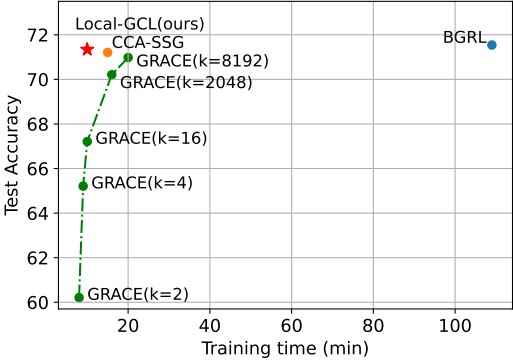

Figure 2: Test accuracy and training time comparison of LOCAL-GCL, BGRL, CCA-SSG and GRACE with subsampling.

LOCAL-GCL' sensitivities with respect to them in Fig. 1. The observations are two-fold: 1) Similar to other self-supervised learning models (Hassani & Ahmadi, 2020; Zhu et al., 2021; Zhang et al., 2021; Lee et al., 2021), LOCAL-GCL that benefit from a large embedding dimension (e.g., 512), LOCAL-GCL's performance also grows as we increase the embedding dimension $d$. However, the optimal embedding dimension for LOCAL-GCL is even larger, e.g., 2048 on Cora and Citeseer (note that other methods do not benefit from such a large embedding dimension). This could be justified by our analysis in Theorem 1, which demonstrates that with a larger embedding dimension, the bound of the prediction error will become lower (because the target eigenvalue $\lambda_{d+1}$ becomes larger). 2) Increasing the projection dimension can lead to better but minimal improvement. This indicates that in practice, LOCAL-GCL can perform quite well without using a huge projection $D$.

## 5.4 SCALABILITY COMPARISON

To evaluate the scalability of LOCAL-GCL on real-world datasets, we compare the total training time and accuracy with state-of-the-art scalable methods BGRL (Thakoor et al., 2021) and CCA-SSG (Zhang et al., 2021) on Arxiv dataset. To further justify the necessity of adopting a large number of negative examples, we additionally adopt GRACE (Zhu et al., 2020b), a powerful non-scalable contrastive model. We use sub-sampling strategy to randomly sample a fixed number of negative examples every epoch so that the GRACE model could be fit into a GPU with 16G memory. In Fig. 2 we plot the model's performance and the corresponding training time of different methods. We can see that compared with scalable methods CCA-SSG and BGRL, LOCAL-GCL achieves comparable performance but with the least training time. We can also observe that although reducing the number of negative examples ($k$ in Fig. 2) can enable the model to be trained much faster, the performance drop is significant, which cannot make up for the efficiency benefit. Furthermore, if we continue to use an even larger number of negative examples, GRACE will soon run out of memory. This result demonstrates that LOCAL-GCL can better balance the training efficiency and model performance.

## 6 CONCLUSIONS

In this paper, we have presented LOCAL-GCL, a simple, light-weighted yet effective contrastive model for self-supervised node representation learning. LOCAL-GCL incorporates two orthogonal techniques to address two key issues in contrastive learning on graph data: reliance on data augmentation and scalability, respectively. The former involves a new definition of positive examples such that the model is free from data augmentation whose design could be more difficult than its counterparts in vision and text, and calls for careful customization for different graph data and tasks. The latter devises an approximated contrastive loss, which reduces the quadratic complexity of traditional contrastive learning to linearity in terms of the graph size. Extensive experimental results on seven public graph benchmarks show its efficiency and efficacy.

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

# A    PROOFS

## A.1    PROOF FOR THEOREM 1

*Proof.* To prove Theorem 1, we first introduce a lemma that shows the optimal representations learned from InfoNCE-like loss function (e.g., Eq. 5) derived from Balestriero & LeCun (2022):

**Lemma 1.** *Denote the eigendecomposition of the normalized graph adjacency matrix by $\tilde{A} = U\Lambda U^\top$, where $U$ and $\Lambda$ are the eigenvectors and eigenvalues of ˜ respectively. Assume node embeddings are free vectors, then a global minimizer of the loss function in Eq. 5 is given by:*

$$Z^* = (U\Lambda^{1/2})_{:,1:d}, \tag{11}$$

*up to permutations of the eigenvector associated with the same eigenvalue. $1:d$ in Eq. 11 indicates the largest $d$ eigenvalues and the associated eigenvectors.*

Besides, HaoChen et al. (2021) has presented the following theoretical guarantee for the model learned with the matrix factorization loss:

**Lemma 2.** *For a graph $\mathcal{G}$ with symmetric normalized graph adjacency matrix and Laplacian matrix $\tilde{A}$ and $\tilde{L}$, let $f^*_{mf} \in \arg\min_{f_{mf}:\mathcal{V}\to\mathbb{R}^d}$ be a minimizer of the matrix factorization loss: $\mathcal{L}_{\mathrm{mf}}(F) = \|(I - \tilde{L} - FF^\top)\|_F^2$, where $F$ is the embedding matrix. Then, for any labeling function $\hat{y}: \mathcal{V} \to [r]$, there exists a linear classifier $B^* \in \mathbb{R}^{d\times c}$ with norm $\|B^*\|_F \leq 1/(1 - \lambda_{d+1})$ such that:*

$$\mathbb{E}_{v\in\mathcal{V}}\left[\|\overrightarrow{y}(v) - B^*f^*(v)\|_2^2\right] \leq \frac{1-\phi}{\lambda_{d+1}}, \tag{12}$$

*where $\phi$ is the graph homophily ratio defined in Eq. 6.*

Then we just need to connect $Z^*$ in Eq. 11 with the optimal solution of the rthe matrix factorization loss $\mathcal{L}_{\mathrm{mf}}(F) = \|I - \tilde{L} - FF^\top\|_F^2 = \|\tilde{A} - FF^\top\|$. According to Eckart–Young–Mirsky theorem, the optimal $FF^\top$ is $FF^\top = U_{:,1:d}\Lambda_{:,1:d}U_{:,1:d}^\top$ As a result, $F^* = Z^* = (U\Lambda^{1/2})_{:,1:d}$ is exactly the global minimizer of $\mathcal{L}_{mf}(F)$. Then the proof is complete.

$\square$

## A.2    PROOF FOR THEOREM 2

*Proof.* To prove Theorem 2, we first introduce the following lemma given by Lemma 1 in (Yu et al., 2016).

**Lemma 3.** *The Random Fourier Estimation $K_{RFF}(x, y)$ is an unbiased estimator of the Gaussian kernel, i.e., $\mathbb{E}_\omega(\psi(x)^\top\psi(y)) = e^{-\|x-y\|_2^2/2\tau}$. Let $z = \|x - y\|_2/\sqrt{\tau}$, then the variance of $K_{RFF}(x, y)$ is $\mathbb{V}_\omega(\psi(x)^\top\psi(y)) = (1 - e^{-z^2})^2/2D$.*

The lemma shows that the Random Fourier Features can achieve an unbiased approximation for the Gaussian kernel with a quantified variance.

Back to our main theorem, we can derive the following probability using the Chebyshev's inequality:

$$\mathbb{P}(\Delta < \frac{1 - \exp(-4/\tau)}{\sqrt{2D\varepsilon}}) \geq 1 - \frac{\mathbb{V}_\omega(\psi(h_i)^\top\psi(h_j)) * 2D\varepsilon}{(1 - \exp(-4/\tau))^2} \tag{13}$$

where $\Delta = |\psi(h_i)^\top\psi(h_j) - \kappa^G(h_i, h_j)|$ denotes the deviation of the kernel approximation. Using the result in Lemma 3, we can further obtain that the RHS of Eq. 13 is

$$1 - \frac{(1 - e^{-z^2})^2 * \varepsilon}{(1 - \exp(-4/\tau))^2} \tag{14}$$

As both $h_i$ and $h_j$ are $\ell_2$-normalized, we have $z^2 = \|h_i - h_j\|_2^2/\tau \leq 4/\tau$, so we can conclude the stated result:

$$\mathbb{P}(\Delta < \frac{1 - \exp(-4/\tau)}{\sqrt{2D\varepsilon}}) \geq 1 - \frac{(1 - e^{-z^2})^2 * \varepsilon}{(1 - \exp(-4/\tau))^2} \geq 1 - \varepsilon \tag{15}$$

$\square$

# B  ADVANTAGES OF LOCAL-GCL OVER PRIOR METHODS

We provide a systematic comparison for the proposed LOCAL-GCL with previous typical methods self-supervised node representation learning, including DGI (Velickovic et al., 2019), MVGRL (Hassani & Ahmadi, 2020), GRACE (Zhu et al., 2020b), GCA (Zhu et al., 2021), BGRL (Thakoor et al., 2021), CCA-SSG (Zhang et al., 2021) and AFGRL (Lee et al., 2021).

## B.1  TOTALLY AUGMENTATION-FREE WITH SINGLE BRANCH

Most of previous self-supervised learning models (Velickovic et al., 2019; Hassani & Ahmadi, 2020; Zhu et al., 2020b; 2021; Thakoor et al., 2021; Zhang et al., 2021) require carefully-defined graph augmentations to obtain positive pairs. However, as noted above, it is hard and costly to design task and dataset-specific graph augmentations. Our method LOCAL-GCL naturally avoids data augmentation by using the first-order neighborhood information of each node. Moreover, different from all previous methods that have two branches of models (double amounts of inputs, hidden variables, outputs, etc.), our approach only requires a single branch, which makes our method light-weighted.

## B.2  PROJECTOR/PREDICTOR/DISCRIMINATOR-FREE

Most of the previous methods require additional components besides the basic encoder for decent empirical results (Zhu et al., 2020b; 2021), to break symmetries to avoid trivial solutions (Thakoor et al., 2021; Lee et al., 2021) or to estimate some score functions in their final objectives (Velickovic et al., 2019; Hassani & Ahmadi, 2020). Compared with them, LOCAL-GCL is much simpler, in both conceptual and practical senses, without using any parameterized model but the basic encoder.

## B.3  THEORETICALLY GROUNDED WITH LINEAR COMPLEXITY

Empirically, InfoNCE-based contrastive methods (Zhu et al., 2020b; 2021) often show better performance than DIM-based methods (Velickovic et al., 2019; Hassani & Ahmadi, 2020) yet suffers from the scalability issue. Recent non-contrastive methods (Thakoor et al., 2021; Lee et al., 2021) could maintain decent performance with linear model complexity through asymmetric structures, while the rationale behind their success still remains unclear (Zhang et al., 2021). As a contrastive model, LOCAL-GCL is theoretically grounded (through maximizing the mutual information between target node's embedding and its neighbor's embeddings), and its kernelized approximation of negative loss enables it to scale to large graphs with linear time and memory complexity with respect to the graph size.

# C  EXPERIMENT DETAILS AND ADDITIONAL EMPIRICAL RESULTS

## C.1  DATASETS

The statistics of the used datasets are presented in Table 4, and brief introduction and settings are as follows:

Cora, Citeseer, Pubmed are three widely used node classification benchmarks (Sen et al., 2008; Namata et al., 2012). Each dataset consists one citation network, where nodes represent papers and edges represent a citation relationship from one node to another. We use the public split, where each class has fixed 20 nodes for training, another fixed 500 nodes and 1000 nodes for validation/test, respectively for evaluation.

Coauthor-CS, Coauthor-Physics are co-authorship graphs based on the Microsoft Academic Graph from the KDD Cup 2016 challenge (Sinha et al., 2015). Nodes are authors that are connected by an edge if they co-authored a paper; node features represent paper keywords for each author's papers, and class labels indicate the most active fields of study for each author. As there is no public split for these datasets, we randomly split the nodes into train/validation/test (10%/10%/80%) sets.

Amazon-Computer, Amazon-Photo are segments of the Amazon co-purchase graph (McAuley et al., 2015), where nodes represent goods, edges indicate that two goods are frequently bought

Table 4: Statistics of benchmark datasets.

| Dataset | #Nodes | #Edges | #Classes | #Features | $\phi$ |
|---|---|---|---|---|---|
| Cora | 2,708 | 10,556 | 7 | 1,433 | 0.810 |
| Citeseer | 3,327 | 9,228 | 6 | 3,703 | 0.736 |
| Pubmed | 19,717 | 88,651 | 3 | 500 | 0.802 |
| Coauthor-CS | 18,333 | 327,576 | 15 | 6,805 | 0.808 |
| Coauthor-Physics | 34,493 | 991,848 | 5 | 8,451 | 0.931 |
| Amazon-Computer | 13,752 | 574,418 | 10 | 767 | 0.777 |
| Amazon-Photo | 7,650 | 287,326 | 8 | 745 | 0.827 |
| Ogbn-Arxiv | 169,343 | 2,332,386 | 40 | 128 | 0.655 |
| Chameleon | 2,277 | 36,101 | 5 | 2,325 | 0.235 |
| Squirrel | 5,201 | 217,073 | 5 | 2,089 | 0.224 |
| Actor | 7,600 | 33,544 | 5 | 931 | 0.219 |

Table 5: Details of hyper-parameters of the experimental results in Table 1

| Dataset | Local-GCL | | | | | | | |
|---|---|---|---|---|---|---|---|---|
| | Encoder | # Steps | # layers | lr | wd | $d$ | $D$ | $\tau$ |
| Cora | GCN | 50 | 2 | 5e-4 | 1e-6 | 2048 | 8192 | 0.5 |
| Citeseer | GCN | 20 | 1 | 1e-3 | 1e-4 | 2048 | 4096 | 0.5 |
| Pubmed | GCN | 50 | 2 | 5e-4 | 0 | 1024 | 8192 | 0.5 |
| Amazon-Computer | GCN | 50 | 2 | 5e-4 | 0 | 1024 | 8192 | 0.8 |
| Amazon-Photo | GCN | 50 | 2 | 5e-4 | 1e-6 | 2048 | 8192 | 0.5 |
| Coauthor-CS | MLP | 50 | 2 | 5e-4 | 1e-4 | 1024 | 4096 | 0.5 |
| Coauthor-Physics | MLP | 50 | 2 | 5e-4 | 0 | 1024 | 4096 | 0.7 |

together; node features are bag-of-words encoded product reviews, and class labels are given by the product category. We also use a 10%/10%/80% split for these two datasets.

For simplicity, we use the processed version of these datasets provided by Deep Graph Library (Wang et al., 2019)[2]. One can easily acquire these datasets using the api provided by DGL.

Ogbn-Arxiv is a directed graph, representing the citation network between all Computer Science (CS) arXiv papers (Hu et al., 2020). Each node is an arXiv paper and each directed edge indicates that one paper cites another one. Each paper comes with a 128-dimensional feature vector obtained by averaging the embeddings of words in its title and abstract. The embeddings of individual words are computed by running the skip-gram over the MAG corpus. All papers are also associated with the year that the corresponding paper was published. We use the official split in our experiments.

Chameleon and Squirrel (Rozemberczki et al., 2021) are Wikipedia networks introduced. Nodes represent web pages, and edges represent hyperlinks between them. Node features represent several informative nouns in the Wikipedia pages. The task is to predict the average daily traffic of the web page.

Actor is the actor-only induced subgraph of the film-director-actor-writer network used in Pei et al. (2020). Each node corresponds to an actor, and the edge between two nodes denotes co-occurrence on the same Wikipedia page. Node features correspond to some keywords in the Wikipedia pages. The task is to classify the nodes into five categories in term of words of actor's Wikipedia.

For Chameleon, Squirrel and Actor, we use the raw data provided by the Geom-GCN (Pei et al., 2020) paper[3], and we use the 10-fold split provided.

## C.2 HYPER-PARAMETERS

We provide detailed hyper-parameters on the seven benchmarks in Table 5. All hyper-parameters are selected through a small grid search, and the search space is provided as follows:

---

[2] https://docs.dgl.ai/en/0.6.x/api/python/dgl.data.html
[3] https://github.com/graphdml-uiuc-jlu/geom-gcn

- Training steps: {20, 50}
- Number of layers: {1, 2}
- Embedding dimension $d$: {512, 1024, 2048}
- Projection dimension $D$: {2048, 4096, 8192}
- Temperature $\tau$: {0.5, 0.6, 0.7, 0.8, 0.9, 1.0}
- Learning rate: {5e-4, 1e-3, 5e-3}
- Weight decay: {0, 1e-6, 1e-4}

### C.3 EXTENSIVE ABLATION STUDIES

We further extend LOCAL-GCL with other designs of positive example construction, negative example selection, and self-supervised objective functions to evaluate the effectiveness of each single component of the proposed method.

#### C.3.1 DIFFERENT STRATEGIES FOR CONSTRUCTING POSITIVE EXAMPLES USING NEIGHBORHOODS

In addition to treats each neighboring node equally as positive examples, we further explore two other positive sampling strategies: 1) computing the similarities between neighboring nodes, and use the most similar one as the positive example (we term it LOCAL-GCL-max); 2) reweighting the importance of different neighboring nodes according to their similarities, where more similar neighboring node should be assigned a larger weight (we term it LOCAL-GCL-weight).

For LOCAL-GCL-max, we first its the nearest neighbor:

$$s_i = \underset{z_j, j \in \mathcal{N}(i)}{\arg\min} \left\| \frac{z_j}{\|z_j\|_2^2} - \frac{z_i}{\|z_i\|_2^2} \right\|^2, \tag{16}$$

and the objective function is

$$\mathcal{L}_{LocalGCL-max} = -\frac{1}{|\mathcal{V}|} \sum_{i=1}^{|\mathcal{V}|} \log \frac{\exp(z_i^\top s_i / \tau)}{\sum_{k \in \mathcal{V}} \exp(z_i^\top z_k / \tau)}. \tag{17}$$

For LocalGCL-weight, we first compute the pair-wise similarities of neighboring nodes:

$$\text{sim}_{ij} = A_{i,j} \cdot \frac{z_i^\top z_j}{\|z_i\|_2^2 \|z_j\|_2^2}. \tag{18}$$

Then, for a target node $i$, it softmax the scores of its neighboring nodes as the weights:

$$w_i(j) = \text{softmax}[\text{sim}_{ij}]_{j \in \mathcal{N}(i)}. \tag{19}$$

The objective function of LocalGCL-weight is consequently defined as

$$\mathcal{L}_{LocalGCL-weight} = -\frac{1}{|\mathcal{V}|} \sum_{i=1}^{|\mathcal{V}|} \log \frac{\sum\limits_{j \in \mathcal{N}(i)} w_i(j) \exp(z_i^\top z_j / \tau) / |\mathcal{N}(i)|}{\sum\limits_{k \in \mathcal{V}} \exp(z_i^\top z_k / \tau)}. \tag{20}$$

The results of LOCAL-GCL-max and LOCAL-GCL-weight, together with the original LOCAL-GCL is presented in Table 6: From this table, we can observe that using the nearest neighboring node as the only one positive example is likely harmful (except on `Citeseer`) to the performance, this should be because this operation can discard a lot of useful information in the local neighborhood. Besides, LOCAL-GCL-weight improves the performance on `Cora` significantly but makes little difference on `Citeseer` and `Pubmed`. This might be because, without supervision, a higher similarity in the latent space does not necessarily indicate a higher probability of sharing the same label. Considering that LOCAL-GCL-weight requires additional computational cost for computing the weights, and the performance of the original LOCAL-GCL is already quite good, we just treat every neighboring node equally in this work.

Table 6: Performance variation when using different strategies of constructing positive examples from neighboring nodes.

|  | Cora | Citeseer | Pubmed |
|---|---|---|---|
| LOCAL-GCL (origin) | $84.5 \pm 0.4$ | $73.6 \pm 0.4$ | $82.1 \pm 0.5$ |
| LOCAL-GCL-max | $83.6 \pm 0.5$ | $73.7 \pm 0.5$ | $81.7 \pm 0.5$ |
| LOCAL-GCL-weight | $84.9 \pm 0.4$ | $73.7 \pm 0.4$ | $82.2 \pm 0.5$ |

Table 7: Test accuracy, training time cost and memory cost when using the positive pair constructing method in this paper compared with using data augmentation for BGRL and CCA-SSG.

|  | Coauthor-CS | | | Coauthor-Physics | | |
|---|---|---|---|---|---|---|
|  | Acc | Time | Memory | Acc | Time | Memory |
| BGRL | $92.59 \pm 0.17$ | 30.4 min | 2.9G | $94.48 \pm 0.08$ | 57.7 min | 5.6G |
| BGRL-Local | $92.98 \pm 0.23$ | 18.2 min | 1.8G | $95.82 \pm 0.11$ | 32.3 min | 3.7G |
| CCA-SSG | $93.31 \pm 0.22$ | 1.1 min | 3.5G | $95.38 \pm 0.06$ | 5.9 min | 7.1G |
| CCA-SSG-Local | $94.08 \pm 0.19$ | 0.7 min | 2.2G | $95.69 \pm 0.09$ | 4.3 min | 4.4G |

### C.3.2 COMBINING WITH NEGATIVE-SAMPLE-FREE SELF-SUPERVISED LEARNING METHODS

using the target node's first-order neighbors as positive examples) could also benefit non-contrastive methods, like BGRL and CCA-SSG. Note that both BGRL and CCA-SSG use the exact same data augmentations methods proposed in GRACE (a contrastive method). As a result, we can simply replace the positive examples in BGRL, and on-diagonal terms in CCA-SSG with the ones used in this paper, thus making them free from data augmentations. In such as a case, the models do not require two-viewed data as inputs, so the size of inputs, intermediate variables, and outputs can be reduced by half. To better demonstrate this point, we further extend our method with BGRL and CCA-SSG.

For BGRL, denote the target embedding of node $i$ by $h_i$ and the corresponding prediction by $z_i$, our BGRL-Local optimizes the following loss function:

$$\mathcal{L}_{BGRL-Local} = -\frac{2}{N} \sum_{i \in \mathcal{V}} \sum_{j \in \mathcal{N}(i)} \frac{z_j \cdot h_i}{\|z_j\|_2^2 \|h_i\|_2^2} \tag{21}$$

For CCA-SSG, denote the embedding of node $i$ by $h_i$, we first compute the local summary of node $i$ by $z_i = \frac{1}{|\mathcal{N}(i)|} \sum_{j \in \mathcal{N}(i)} h_j$. Then we compute two standardized embedding matrix by:

$$\tilde{H} = \frac{H - \mu(H)}{\sigma(H) * \sqrt{|\mathcal{V}|}}, \tilde{Z} = \frac{Z - \mu(Z)}{\sigma(Z) * \sqrt{|\mathcal{V}|}}. \tag{22}$$

Finally, CCA-SSG-Local optimizes the following loss function:

$$\mathcal{L}_{CCA-SSG-Local} = \left\| \tilde{H} - \tilde{Z} \right\|_F^2 + \lambda \left( \left\| \tilde{H}^\top \tilde{H} - I \right\|_F^2 + \left\| \tilde{Z}^\top \tilde{Z} - I \right\|_F^2 \right). \tag{23}$$

We report the performance of BGRL-Local and CCA-SSG-Local compared with the original BGRL and CCA-SSG on CS and Physics, together with their training time/memory cost in Table 7 (We didn't choose the three citation networks as BGRL performs really bad on them).

(BGRL takes much longer time for training, because it does require a lot of training epochs to converge.) As demonstrated in the table. The performance of BGRL and CCA-SSG both get improved after adopting neighboring nodes as positive examples, which highlights the value of graph structure information as self-supervised learning signals. Besides, it also reduces the training time and memory cost because it gets rid of the two-branch architecture caused by data augmentations.

### C.3.3 COMBINING WITH HARD NEGATIVE SAMPLING METHODS

Furthermore, one important advantage of contrastive learning is that it can be incorporated with hard negative sampling techniques (Robinson et al., 2021; Kalantidis et al., 2020; Zhang et al.,

Table 8: Effect of combining LOCAL-GCL with hard-negative sampling methods.

|  | Cora | Citeseer | Pubmed |
|---|---|---|---|
| LOCAL-GCL (origin) | $84.5 \pm 0.4$ | $73.6 \pm 0.4$ | $82.1 \pm 0.5$ |
| LOCAL-GCL +ProGCL-mix | $84.9 \pm 0.5$ | $74.2 \pm 0.4$ | $82.3 \pm 0.5$ |

Table 9: Bootstrapped confidence interval (with 95% confidence).

|  | Cora | Citeseer | Pubmed | Computer | Photo | CS | Physics |
|---|---|---|---|---|---|---|---|
| CCA-SSG | [84.06, 84.30] | [73.05, 73.21] | [81.51, 81.72] | [88.57, 88.86] | [93.10, 93.22] | [93.25, 93.41] | [95.37, 95.41] |
| LOCAL-GCL | [84.43, 84.75] | [73.45, 73.83] | [81.82, 82.29] | [88.66, 89.06] | [93.20, 93.48] | [94.89, 94.98] | [96.26, 96.35] |

2022; Xia et al., 2022). These works either assign weights to different negative samples so that true negative examples are more important than false negative examples, or use mix-up methods to generate even harder negative examples. With these techniques, the performance of contrastive learning can be greatly boosted. To demonstrate this point, we further combine our method with the hard negative mining strategy ProGCL-mix proposed in Xia et al. (2022), and we report the performance comparison on Cora, Citeseer and Pubmed in Table 8: The improvement is prominent (especially on Citeseer) thanks to the hard-negative sampling strategy(however, ProGCL-mix is not our contribution so we cannot include it in our method in this paper). Besides, we notice that the improvement on Pubmed is smaller than that on the other two datasets, we guess this is because there are only 3 classes of nodes on Pubmed, so it is less effective to mine hard negatives than 7 classes on Cora and 6 on Citeseer. By contrast, negative-sample-free method like BGRL and CCA-SSG cannot get benefits from it.

### C.4 ANALYSIS OF THE RESULTS ON HETEROPHILIC GRAPHS

Although a little bit counter-intuitive, the better performance of our method on heterophilic graphs does exist, and can be justified through the differences between the two distinguished methods of generating positive pairs: 1) data augmentations on graph data; 2) two connected nodes being a positive pair. Two recent studies Wang et al. (2022) and Liu et al. (2022) both demonstrate that existing data augmentations on graphs, such as graph diffusion used in MVGRL (Hassani & Ahmadi, 2020), edge dropping and feature masking used in GRACE (Zhu et al., 2020b), GCA (Zhu et al., 2021), BGRL (Thakoor et al., 2021) and CCA-SSG (Zhang et al., 2021) have larger impacts on the high-frequency information of the graph than the low-frequency information. As a result, the differences between two graph views are about low-frequency components. Considering that contrastive learning aims at maximizing the mutual information shared between the two views, only the invariant information (low-frequency) is encouraged to be learned by the embeddings, while the middle and high-frequency information is discarded.

On the other hand, studies on GNNs for heterophilic graphs (Bo et al., 2021) demonstrate that for heterophilic graphs, the high-frequency information in the graph is more effective for downstream classification performance. Considering that data-augmentation-based methods focus on low-frequency information while neglecting high-frequency one, it is not strange that they cannot give satisfactory performance. By contrast, as high-frequency knowledge in the graph represents the differences between the node feature with its neighborhood features, the edge-wise contrastive loss used in our work can better capture the differences in the information between two connected nodes, thus performing better on heterophilic graphs.

### C.5 STATISTICAL SIGNIFICANCE TEST

We further conduct statistical significance tests to show that LOCAL-GCL does perform better than the previous SOTA method CCA-SSG (Zhang et al., 2021).

In Table 9 we present the bootstrapped confidence interval (with 95% confidence). We also conduct Wilcoxon signed-rank test on the proposed LOCAL-GCL and the second best model CCA-SSG, using the averaged test accuracy on the 7 datasets in Table 1. The p-value of CCA-SSG and LOCAL-GCL's performance on the 7 datasets is $0.0325 < 0.05$. As a result, we can conclude that there is a genuine performance difference between the performance of LOCAL-GCL and that of CCA-SSG.

# D REPRODUCIBILITY

We provide detailed codes and running instructions in the supplementary material.

