# OpenReview forum: "Localized Graph Contrastive Learning"
_ICLR.cc/2023/Conference — Submitted to ICLR 2023_

### Official Review · Reviewer_BqrR · 2022-10-24

**Confidence:** 5
**Correctness:** 2
**Technical Novelty And Significance:** 2
**Empirical Novelty And Significance:** 2
**Recommendation:** 5

**Clarity, Quality, Novelty And Reproducibility:**

The paper makes a good try to reduce the computation bottleneck of the current graph CL models. But the theory and claim in the paper are not well-supported.

**Strength And Weaknesses:**

The following are the pros of the paper:
1. The paper proposes a new graph contrastive learning without relying on any hand-crafted graph augmentation techniques. According to the reported results, the proposed method achieves SOTA performances on many datasets.

2. The paper design a surrogate loss to approximate the negative loss in contrastive learning and thus reduce the quadratic computation cost in linear complexity. Experiments are conducted to demonstrate the obvious complexity advantages of the proposed method over prior graph CL models with extensive negative sampling (GRACE etc.)

3. Overall, the paper is well-organized and easy to read.

The following are some concerns and questions about the paper:
1. Although the proposed model can achieve better computation efficiency compared with those GCL methods with extensive negative sampling, this advantage does not exist for those negative-sample-free methods, e.g., BGRL and CCA-SSG. Besides, the performance gain over the two baselines seems to be incremental, the authors are encouraged to clarify the contribution of the proposed methods over the prior negative-sample-free graph CL methods.

2. The proposed augmentation-free mechanism is designed based on the graph homophily theory. According to the assumption and theoretical justification of the paper, the performance of the proposed method should be heavily dependent on the homophily ratio. However, the performance gain of the method seems to be even more significant on the heterophily graphs, which conflicts with the paper’s assumption. It is very important to conduct more discussions and experiments to study this phenomenon because this is the most important claim of this paper. The authors are encouraged to provide more explanation on the phenomenon and testify to the correlation between its work mechanism and graph homophily.

3. It is good to see the authors try to theoretically justify their claims in the paper, but the proof of theorem 1 is still confusing. For example, how do you derive Equation (14) from Equation 12 and how do you derive the relation between the homophily ratio and the node embeddings that minimize matrix factorization loss? The authors are encouraged to provide more detailed proof of their proposed theorem.

4. The ablation study in the paper can be only considered as the parameter sensitivity analysis. A more comprehensive ablation study is necessary to evaluate the design of the paper, for example, what the performance and scalability of the proposed method will be with different positive and negative sampling strategies. Besides, the paper only compares the training time of the proposed methods and selected baselines, it would be better if the experiment about training space comparison is included.

**Summary Of The Paper:**

The paper proposes an augmentation-free graph contrastive learning framework based on the assumption of the graph homophily assumption. Meanwhile, the authors use the Random Fourier Features to approximate the negative loss in contrastive learning and thereby reduce the computation cost to linear complexity.

**Summary Of The Review:**

Overall, the idea is interesting and efficient, but the contribution of the idea over the prior negative -sample-free graph SSL method is limited. More importantly, the proposed claim is not well-supported, and the authors do not provide a very detailed analysis of the theorem in the paper.  Except for the performance comparison, the paper does not demonstrate enough experiments to support the claim and show the superiority of the proposed method over the baselines.  Additional theoretical analysis and experiments mentioned above are necessary to strengthen this work.

---

> ### Author Response · Authors · 2022-11-16
> **Respones to Reviewer BqrR, batch 1 of 4**
>
> We thank the reviewer for providing such thorough comments and the valuable suggestions that will be a great help for improving this work. Here we'd like to address your concerns:
>
> Q1: Contributions of our work to negative-free methods.
>
> A1: In terms of the background, we'd like to clarify the focus of this work and the concept of **contrastive learning**. As stated in the title and abstract, one purpose of of our work is to mitigate the inefficiency problem of contrastive learning (i.e., InfoNCE loss) on graphs, so we didn't pay much attention to non-contrastive methods in our submission. Please note that although both BGRL and CCA-SSG are two-branched (multi-view) self-supervised learning methods, they are non-contrastive in that they do not contrast between positive pairs and negative pairs (although these two concepts are confused in some works).
>
>
> Second, besides the approximation of the negative loss, our first contribution (i.e., using the target node's first-order neighbors as positive examples) could also benefit non-contrastive methods, like BGRL and CCA-SSG. Note that both BGRL and CCA-SSG uses the exactly same data augmentations methods proposed in GRACE (a contrastive method). As a result, we can simply replace the positive examples in BGRL, and on-diagonal terms in CCA-SSG with the ones used in this paper, thus getting them free from data augmentations. In such as a case, the models does not require two-viewed data as inputs, so the size of inputs, intermediate variables and outputs can be reduced by half. To better demonstrate this point, we further extends our method with BGRL and CCA-SSG.
>
> For BGRL, denote the target embedding of node $i$ by $\mathbf{h}_i$ and the corresponding prediction by $\mathbf{z}_i$, our BGRL-Local optimizes the following loss function:
>
> $$
>    \mathcal{L}_{BGRL-Local} = -\frac{2}{N} \sum\limits\_{i \in \mathcal{V}} \sum\limits\_{j \in \mathcal{N}(i)} \frac{\mathbf{z}_j \cdot \mathbf{h}_i}{\Vert \mathbf{z}_j \Vert_2^2  \Vert \mathbf{h}_i \Vert_2^2}
> $$
>
> For CCA-SSG, denote the embedding of node $i$ by $\mathbf{h}_i$,
>  we first compute the local summary of node $i$ by $\mathbf{z}_i = \frac{1}{|\mathcal{N}(i)|} \sum\limits\_{j\in\mathcal{N}(i)}
>  \mathbf{h}_j$. Then we compute two standardized embedding matrix by:
>
>  $$
>     \tilde{\mathbf{H}} = \frac{\mathbf{H} -\mu(\mathbf{H})}{\sigma(\mathbf{H})*\sqrt{|\mathcal{V}|}},  \tilde{\mathbf{Z}} = \frac{\mathbf{Z} -\mu(\mathbf{Z})}{\sigma(\mathbf{Z})*\sqrt{|\mathcal{V}|}}.
>  $$
>
> Finally, CCA-SSG-Local optimizes the following loss function:
>
> $$
>   \mathcal{L}_{CCA-SSG-Local} = \left\| \tilde{\mathbf{H}} - \tilde{\mathbf{Z}} \right\|_F^2 + \lambda \left( \left\|  \tilde{\mathbf{H}}^{\top}\tilde{\mathbf{H}} - \mathbf{I} \right\|_F^2 + \left\|  \tilde{\mathbf{Z}}^{\top}\tilde{\mathbf{Z}} - \mathbf{I} \right\|_F^2 \right).
> $$
>
> We report the performance of BGRL-Local and CCA-SSG-Local compared with the original BGRL and CCA-SSG on CS and Physics, together with their training time/memory cost in the following table (We didn't choose the three citation networks as BGRL performs really bad on them):
>
>
> |      | CS  | CS  | CS  |  Physics | Physics  | Physics |
> | ---- | ---- | ----  |  ----   | ----  |  ----   | ----  |
> |      | Acc  | Time  | Memory  |  Acc | Time  | Memory |
> | BGRL  | 92.59 $\pm$ 0.17 | 30.4 min  |  2.9G  | 95.48 $\pm$ 0.08 | 57.7 min  | 5.6G |
> | BGRL-Local  | 92.98 $\pm$ 0.23 | 18.2 min  | 1.8G | 95.82 $\pm$ 0.11| 32.3 min  | 3.7G |
> | CCA-SSG  | 93.31 $\pm$ 0.22 |  1.1 min  | 3.5G | 95.38 $\pm$ 0.06 |  5.9 min  | 7.1G |
> | CCA-SSG-Local  | 94.08 $\pm$ 0.19 | 0.7 min  | 2.2G | 95.69 $\pm$ 0.09 |  4.3 min  | 4.4G |
>
> (BGRL takes much longer time for training, because it does requires a lot of training epochs to converge.) As demonstrated in the table. The performance of BGRL and CCA-SSG both get improved after adopting neighboring nodes as positive examples, which highlights the value of graph structure information as self-supervised learning signals. Besides, it also reduces the training time and memory cost because it gets rid of the two-branch architecture caused by data augmentations.

---

> > ### Author Response · Authors · 2022-11-16
> > **Respones to Reviewer BqrR, batch 1 of 4 (following)**
> >
> > Third, even though negative-free methods like BGRL and CCA-SSG are much more efficient than the vanilla contrastive methods with negative samples, and can demonstrate competitive performance, contrastive methods are still important and have incomparable advantages than non-contrastive ones. Besides, the theoretical guarantees (e.g., lower bound of mutual information [R1], and the performance guarantee (Theorem 1)). Another typical example is hard negative sampling techniques [R2, R3, R4, R5]. These works either assign weights to different negative samples so that true negative examples are more important than false negative examples, or use mix-up methods to generate even harder negative examples. With these techniques, the performance of contrastive learning can be greatly boosted. To demonstrate this point, we further combine our method with the hard negative mining strategy ProGCL-mix proposed in [R5], and we report the performance comparison on Cora, Citeseer and Pubmed in the following table:
> >
> > |  | Cora   | Citeseer  | Pubmed |
> > |  ----   | ----  | ---- | ----  |
> > | Local-GCL(origin)  | 84.5 $\pm$ 0.4 | 73.6 $\pm$ 0.4 |82.1 $\pm$ 0.5|
> > | Local-GCL + ProGCL-mix  | 84.9 $\pm$ 0.5 | 74.2 $\pm$ 0.4 |82.3 $\pm$ 0.5|
> >
> > The improvement is prominent (especially on Citeseer) thanks to the hard-negative sampling strategy (however, ProGCL-mix is not our contribution so we cannot include it in our method in this paper). Besides, we notice that the improvement on Pubmed is smaller than that on the other two datasets, we guess this is because there are only 3 classes of nodes on Pubmed, so it is less effective to mine hard negatives than 7 classes on Cora and 6 on Citeseer. By contrast, negative-sample-free method like BGRL and CCA-SSG cannot get benefits from it.
> >
> > Last but not least, we want to emphasize that the improvement is not is not marginal, considering that as an **unsupervised** method, our model's performance is already very close to SOTA **(semi)supervised** GNN methods with complicated architectures and regularizations. For example, one of the SOTAs on Cora and Pubmed is GRAND [R6], which employs a high-order convolution model, combined with consistency regularization between embeddings of different views generated from random data augmentations. GRADN achieves 85.4\% on Cora and 82.7\% on Pubmed, which are less than $1\%$ percent improvement than our results. Considering the existence of such an upper bound, our model's improvement is not marginal.
> >
> > References
> >
> > [R1] Poole, Ben, et al. "On variational bounds of mutual information." International Conference on Machine Learning. PMLR, 2019.
> >
> > [R2] Robinson, Joshua, Ching-Yao Chuang, Suvrit Sra, and Stefanie Jegelka. "Contrastive learning with hard negative samples." arXiv preprint arXiv:2010.04592 (2020).
> >
> > [R3] Kalantidis, Yannis, Mert Bulent Sariyildiz, Noe Pion, Philippe Weinzaepfel, and Diane Larlus. "Hard negative mixing for contrastive learning." Advances in Neural Information Processing Systems 33 (2020): 21798-21809.
> >
> > [R4] Zhang, Shaofeng, Meng Liu, Junchi Yan, Hengrui Zhang, Lingxiao Huang, Xiaokang Yang, and Pinyan Lu. "M-Mix: Generating Hard Negatives via Multi-sample Mixing for Contrastive Learning." In Proceedings of the 28th ACM SIGKDD Conference on Knowledge Discovery and Data Mining, pp. 2461-2470. 2022.
> >
> > [R5] Xia, Jun, Lirong Wu, Ge Wang, Jintao Chen, and Stan Z. Li. "Progcl: Rethinking hard negative mining in graph contrastive learning." In International Conference on Machine Learning, pp. 24332-24346. PMLR, 2022.
> >
> > [R6] Feng, Wenzheng, et al. "Graph random neural networks for semi-supervised learning on graphs." Advances in neural information processing systems 33 (2020): 22092-22103.

---

> ### Author Response · Authors · 2022-11-16
> **Respones to Reviewer BqrR, batch 2 of 4**
>
> Q2: More explanations about the results on heterophily graphs.
>
> A2: Although a little bit counter-intuitive, the better performance of our method on heterophilic graphs does exist, and can be justified through the differences between the two distinguished methods of generating positive pairs: 1) data augmentations on graph data; 2) two connected nodes being a positive pair. Two recent studies [R7] and [R8] both demonstrate that existing data augmentations on graphs, such as graph diffusion used in MVGRL, edge dropping and feature masking used in GRACE, GCA, BGRL and CCA-SSG have larger impacts on the high-frequency information of the graph than the low-frequency information. As a result, the differences between two graph views are about low-frequency components. Considering that contrastive learning aims at maximizing the mutual information shared between the two views, only the invariant information (low-frequency) is encouraged to be learned by the embeddings, while the middle and high frequency information is discarded.
>
> On the other hand, studies on GNNs for heterophilic graphs [R9] demonstrate that for heterophilic graphs, the high-frequency information in the graph is more effective for downstream classification performance. Considering that data augmentation based methods focus on low-frequency information while neglecting high-frequency one, it is not strange that they cannot give satisfactory performance. By contrast, as high frequency knowledge in the graph represents the differences between the node feature with its neighborhood features, the edge-wise contrastive loss used in our work can better capture the differences in the information between two connected nodes, thus performing better on heterophilic graphs.
>
> We've added the detailed explanations for the performance on heterophilic graphs in the revised version.
>
> References:
>
> [R7] Wang, Haonan, et al. "Augmentation-Free Graph Contrastive Learning." arXiv preprint arXiv:2204.04874 (2022).
>
> [R8] Liu, Nian, et al. "Revisiting Graph Contrastive Learning from the Perspective of Graph Spectrum." arXiv preprint arXiv:2210.02330 (2022).
>
> [R9] Bo, Deyu, et al. "Beyond low-frequency information in graph convolutional networks." Proceedings of the AAAI Conference on Artificial Intelligence. Vol. 35. No. 5. 2021.

---

> ### Author Response · Authors · 2022-11-16
> **Respones to Reviewer BqrR, batch 3 of 4**
>
> Q3: Proof for Theorem 1 is not clear.
>
> A3:
> The relation between homophily ratio $\phi$ and node embeddings learned from matrix factorization loss is a direct conclusion of Lemma 2, which is initially presented by Theorem B.3 of [R7]. The whole proof of Theorem B.3 of [R7] is complicated, which consists of a series of Lemmas. As a result, it is inappropriate to present its proof in our paper. Since the reviewer would like us to present more details about the proof of it, here we present the major steps that build a connection between graph homophily ratio $\phi$, the eigenvalues/eigenvectors of the normalized graph Laplacian matrix, and a specific labeling function that induces the homophily ratio.
>
> First, we have to introduce some notations. For a graph $\mathcal{G} = (\mathcal{V}, \mathcal{E})$ with $|\mathcal{V}|$ nodes, the normalized Laplacian matrix is $\mathcal{L}$. $v_i$ is the i-th smallest unit-norm eigenvector of $\mathcal{L}$ with eigenvalue $\lambda_i$.
>
> Step 1 is to show that for any vector $u \in \mathbb{R}^{|\mathcal{V}|}$,
> there exists a linear combination the smallest $d$ eigenvectors such that the error of the estimation of $u$ is bounded by the $d+1$ smallest eigenvalue. Formally,
>
> $$
>   \Vert u - \sum\limits\_{i=1}^d b_iv_i  \Vert_2^2 \le \frac{R(u)}{\lambda\_{d+1}} \Vert u \Vert_2^2,
> $$
> where each $b_i$ is a scalar, $R(u) = \frac{u^{\top}\mathcal{L}u}{u^{\top}u}$ is the Rayleigh quotient of the vector $u$.
>
> The proof is trivial: the vector $u$ can always be decomposed into the summation of the eigenvector basis:
>
> $$
>     u = \sum\limits_{i=1}^{\mathcal{V}} \zeta_i v_i.
> $$
> Besides, we have:
> $$
>     R(u) = \frac{\sum_{i=1}^{\mathcal{V}} \lambda_i \zeta_i^2}{\Vert u \Vert_2^2}.
> $$
>
> As a result, we can simply let $b_i = \zeta_i$. Also, as $\Vert b \Vert_2^2 \le \Vert u \Vert_2^2$, we can conclude that:
>
> $$
>   \Vert u - \sum\limits\_{i=1}^d b_iv_i  \Vert_2^2 = \sum\limits\_{i=k+1}^{|\mathcal{V}|} \zeta_i^2 \le \frac{R(u)}{\lambda_{d+1}} \Vert u \Vert_2^2
> $$
>
> The second step is to show that when the $u$ is defined by an extended labeling function $\hat{y}$, i.e., $u^{\hat{y}}_i(v) = \mathbb{I}[\hat{y}(v) = i]$ ($i$ denotes the i-th class) which gives a class-level graph homophily ratio $\phi^{\hat{y}}_i$, the Rayleigh quotient of the vector $u_i^{\hat{y}}$ is directly related to the graph homophily ratio $\phi^{\hat{y}}_i$:
> $$
>    R(u^{\hat{y}}_i) = \frac{1}{2} (1-\phi^{\hat{y}}_i).
> $$
>
> To prove this, note that $u_i^{\hat{y}}$ projects a node in the graph to an scalar (either 0 or 1), then:
>
> $$
>     u^{\top}\mathcal{L}u =  \Vert u \Vert_2^2 - u^{\top}D^{-1/2}AD^{-1/2}u  \\
>     = \sum\limits_{v \in \mathcal{V}} u(v)^2 - \sum\limits_{(v, v') \in \mathcal{E}} (u(v) - u(v'))^2 \\  = \frac{1}{2} \sum\limits_{(v, v') \in \mathcal{E}} (u(v) - u(v'))^2.
> $$
>
> As a result, we have
>
> $$
>     R(u_i^{\hat{y}}) = \frac{{u_i^{\hat{y}}}^{\top}\mathcal{L}u_i^{\hat{y}}}{{u_i^{\hat{y}}}^{\top} u_i^{\hat{y}}} \\ = \frac{1}{2} \frac{\sum\limits\_{(v,v')\in \mathcal{E}} (u_i^{\hat{y}}(v)-u_i^{\hat{y}}(v'))^2}{\sum\limits\_{v\in\mathcal{V}} u_i^{\hat{y}}(v)^2}.
> $$
> Setting $u_i^{\hat{y}}(v) = 1$ when $v$ is labeled as $i$, and $0$ otherwise, then we can find that the last term is exactly the probability that when one node is labeled as $i$, while its neighbors is not, i.e. $1-\phi^{\hat{y}}_i$. As a result, we complete the proof.
>
> Combining Step 1, and Step 2, we are able to build a connection between a labeling function, the induced graph homophily ratio and the eigenvalues/eigenvectors of the normalized graph Laplacian matrix. For more detailed proof, we recommend the reviewer to directly read [R7] if you are interested.
>
> For the derivation from Eq.(12) to Eq.(14). From Lemma 1, we already have $\mathbf{Z}^* = (\mathbf{U}\Lambda^{1/2})_{:, 1:d}$, where $1:d$ indicates the largest $d$ eigenvalues and the associated eigenvectors.
>
> Then, for the matrix factorization loss
>
> $\mathcal{L}_{\text{mf}}(\mathbf{Z}) = \Vert \mathbf{I} - \tilde{\mathbf{L}} - \mathbf{Z}\mathbf{Z}^{\top} \Vert_F^2 = \Vert \tilde{\mathbf{A}} - \mathbf{Z}\mathbf{Z}^{\top} \Vert$,
>
>  the optimal
> $\mathbf{Z}\mathbf{Z}^{\top}$ is
>
>  $\mathbf{Z}\mathbf{Z}^{\top} = \mathbf{U}\_{:, 1:d}\Lambda\_{:, 1:d}\mathbf{U}\_{:, 1:d}^{\top}$
>
>  according to Eckart–Young–Mirsky theorem. As a result.
>  $\mathbf{Z}^* = (\mathbf{U}\Lambda^{1/2})\_{:, 1:d}$ is exactly the global minimizer of $\mathcal{L}\_{mf}(\mathbf{Z})$. Then the proof is complete.
>
> We've updated the proof in the revised version to make it more clear.
>
> References:
>
> [R7] Jeff Z HaoChen, Colin Wei, Adrien Gaidon, and Tengyu Ma. Provable guarantees for self-supervised
> deep learning with spectral contrastive loss. NeurIPS, 34:5000–5011, 2021.

---

> ### Author Response · Authors · 2022-11-16
> **Respones to Reviewer BqrR, batch 4 of 4**
>
> Q4: More comprehensive ablation studies.
>
> A4: In the response of Q1, we have included two more ablation studies: 1) extending the positive sampling strategies to negative-sample-free methods like BGRL and CCA-SSG; 2) adopt a advanced negative sampling strategy ProGCL-mix. Both experimental results demonstrate the proposed positive sampling method and approximated loss can be successfully combined with other techniques.
>
> In addition to treats each neighboring node equally as positive examples, we further explore two other positive sampling strategies: 1) compute the similarities between neighboring nodes, and use the most similar one as the positive example (we term it LocalGCL-max); 2) reweight the importance of different neighboring nodes according to their similarities, where more similar neighboring node should be assigned a larger weight (we term it LocalGCL-weight).
>
> For LocalGCL-max, we first its the nearest neighbor:
> $$
>     \mathbf{s}\_i = \mathop{\arg\min}_{\mathbf{z}_j, j \in \mathcal{N}(i)} \Vert \frac{\mathbf{z}_j}{\Vert \mathbf{z}_j \Vert_2^2} - \frac{\mathbf{z}_i}{\Vert \mathbf{z}_i \Vert_2^2} \Vert^2,
> $$
>
> and the objective function is
>
> $$
>     \mathcal{L}_{LocalGCL-max} = -\frac{1}{|\mathcal{V}|} \sum\limits\_{i=1}^N  \log \frac{  \exp (\mathbf{z}_i^{\top}\mathbf{s}_i/\tau)}{\sum\limits\_{k \in \mathcal{V}} \exp (\mathbf{z}_i^{\top}\mathbf{z}_k /\tau)} .
> $$
>
> For LocalGCL-weight, we first compute the pair-wise similarities of neighboring nodes:
>
> $$
>     \text{sim}\_{ij} =  \mathbf{A}\_{i,j} \cdot  \frac{\mathbf{z}_i^{\top} \mathbf{z}\_{j}}{\Vert \mathbf{z}_i \Vert_2^2 \Vert \mathbf{z}_j \Vert_2^2}
> $$
>
> Then, for a target node $i$, it softmax the scores of its neighboring nodes as the weights:
>
> $$
>     w_{i}(j) = \text{softmax} [\text{sim}_{ij}]\_{j \in \mathcal{N}(i)}.
> $$
>
> The objective function of LocalGCL-weight is consequently defined as
>
> $$
>     \mathcal{L}_{LocalGCL-weight} = -\frac{1}{|\mathcal{V}|} \sum\limits\_{i=1}^N  \log \frac{\sum\limits\_{j \in \mathcal{N}(i)}  w_i(j)\exp (\mathbf{z}_i^{\top}\mathbf{z}_j/\tau) / |\mathcal{N}(i)|}{\sum\limits\_{k \in \mathcal{V}} \exp (\mathbf{z}_i^{\top}\mathbf{z}_k /\tau)}.
> $$
>
> The results of LocalGCL-max and LocalGCL-weight, together with the original LocalGCL is presented in the following table:
>
> |  | Cora   | Citeseer  | Pubmed |
> |  ----   | ----  | ---- | ----  |
> | LocalGCL(origin)  | 84.5 $\pm$ 0.4 | 73.6 $\pm$ 0.4 |82.1 $\pm$ 0.5|
> | LocalGCL-max  | 83.6 $\pm$ 0.5 | 73.7 $\pm$ 0.5 |81.7 $\pm$ 0.5|
> | LocalGCL-weight  | 84.9 $\pm$ 0.4 | 73.7 $\pm$ 0.4 |82.2 $\pm$ 0.5|
>
> From this table, we can observe that using the nearest neighboring node as the only one positive example is likely harmful (except on Citeseer) to the performance, this should be because this operation can discard a lot of useful information in the local neighborhood. Besides, LocalGCL-weight improves the performance on Cora significantly but makes little difference on Citeseer and Pubmed. This might be because, without supervision, a higher similarity in the latent space does not necessarily indicate a higher probability of sharing the same label. Considering that LocalGCL-weight requires additional computational cost for computing the weights, and the performance of the original LocalGCL is already quite good, we just treat every neighboring node equally in this work.
>
> All the three ablation studies have been added in the revised version in Appendix.

---

> ### Comment · Reviewer_BqrR · 2022-11-19
> **Reply to authors' Response**
>
> Thanks for the detailed explanation part and additional experiments, which make the paper easier to follow. I think the authors make an interesting try to propose a new method to generate positive pairs without relying on traditional augmentation techniques. The paper still has some problems.
>
> First, computation efficiency is one of the most important contributions claimed by the paper, but the proposed method only achieves incremental performance gain compared with other more efficient graph self-supervised learning methods (i.e., BGRL and CCA-SSG) according to the experimental results reported in Q1. More importantly, the proposed method is outperformed by the two baselines on the large-scale datasets, which undermines the contribution of the paper.
>
> Second, the assumption of the paper is not well-supported by its theoretical analysis and experiments. Although the authors cite a few previous papers to justify this unexpected experimental result in Q2, the graph homophily ratio never serves as the assumption or guides the model design in those papers. The working mechanism of GNN is to model the neighborhood distribution of different nodes instead of directly aligning the distance between the connected nodes. Therefore, I suppose the working mechanism of the proposed method is not as the paper assumed. The authors are encouraged to rethink the assumption and rationality behind it.

---

> > ### Author Response · Authors · 2022-11-23
> > **Responses to the follow-up questions.**
> >
> > We thank the reviewer for the follow-up feedback and nice suggestions that can help for improvement. However, we believe there exist some misunderstandings about our work, which we'd like to clarify in detail below.
> >
> > **Efficiency and Performance Gain**
> >
> > The reviewer raised some concerns about the limited performance gain, which might stem from some misunderstandings regarding our central claims. As illustrated in Sec. 1 of our paper, our central claims on the empirical side lie in two-folds:
> >
> > - ***Claim 1***: We propose a new model to enhance the efficiency of contrastive learning models on graph data.
> > - ***Claim 2***: We identify a simple yet effective way for constructing positive pairs, which brings up overall competitive and sometimes even superior performance than other more sophisticated designs.
> >
> > >  "the proposed method only achieves incremental performance gain compared with other more efficient graph self-supervised learning methods (i.e., BGRL and CCA-SSG)"
> >
> > According to Claim 1, our method is mainly designed for improving the efficiency of *contrastive learning* models, while the mentioned methods BGRL and CCA-SSG belong to *non-contrastive learning* models. Our achieved results indeed support our Claim 1. Compared with the previous SOTA contrastive learning method GCA (GCA is an enhanced version of GRACE by using adaptive data augmentations), the proposed Local-GCL significantly reduces the training time and memory cost, by up to 60% and 99%, respectively. Specifically, on Coauthor-CS dataset with $34,493$ nodes, GCA takes 23.7G memory and 95 minutes for training. By contrast, Local-GCL only takes 8.3G and 40s.
> >
> > As an extension, we further compare with non-contrastive methods BGRL and CCA-SSG, and show our proposed designs can also boost their efficiency as a by-product of our model. As shown in our new experiments presented in Table 7 in the revised paper, when equipped with the positive examples of Local-GCL, the improved versions BGRL-Local and CCA-SSG-Local reduce approximately half time/memory costs than their original counterpart BGRL and CCA-SSG. Apart from better efficiency, our design even brings up accuracy gain over them. With our main contributions supported by our achieved results in comparison with contrastive learning models, this extended discussion/result on non-contrastive models further strengthens our contribution.
> >
> > > "the proposed method is outperformed by the two baselines on the large-scale datasets, which undermines the contribution of the paper."
> >
> > According to Claim 2 described at the beginning, the central goal of our model is not aimed at achieving superior accuracy in all the cases but to show the efficacy and competitiveness of a much simpler design for positive pair construction across various datasets. On the mentioned two large-scale datasets, one can see from Table 2 that all the models, including ours yield comparable performance, while critically, our model costs 40\% time/memory than the most efficient competitor CCA-SSG (see Fig.2). Despite this, one step further, our simple model even outperforms previous SOTA models in six out of seven datasets shown in Table 1. We believe these results are strong enough as our contributions.
> >
> > Moreover, while BGRL and CCA-SSG (slightly) outperform our model in two large-scale datasets at the sacrifice of more time/space costs, our simpler model achieves considerable performance gains over them in many other cases. For example, on the difficult benchmark setting, as shown in Table 7 in the revised version, BGRL-Local improves BGRL by 0.39 on Coauthor-CS and 0.34 on Coauthor-Physics, CCA-SSG-Local improves CCA-SSG by 0.77 on Coauthor-CS and 0.31 on Coauthor-Physics. These improvements are significant compared with the improvement of BGRL/CCA-SSG themselves over the suboptimal baselines GCA/MVGRL/AFGRL etc. Besides, as shown in Table 1, the improvement of our Local-GCL compared with BGRL and CCA-SSG is even larger on Coauthor-CS and Coauthor-Physics: Local-GCL improves the performance on Coauthor-CS by over 1.6 and Coauthor-Physics by over 0.8.

---

> > > ### Author Response · Authors · 2022-11-23
> > > **Follow-up**
> > >
> > > **Assumptions not well supported**
> > >
> > > Regarding the graph homophily, we'd like to clarify upfront that although our method is motivated by the homophily phenomenon, the homophily property of graphs is **not** a pre-requisite for our method, and the design of our model does not rely on the homophily ratio $\phi$, which suggests its generality and wide applicability. That being said, we also discuss how the homophily ratio of input graphs could impact the model performance through our theoretical analysis and experiments.
> > >
> > > In Theorem 1, we give an upper bound of the linear classification error w.r.t. $\phi$, which indicates that larger $\phi$ contributes to a tighter bound. However, it doesn't necessarily mean the performance on non-homophilic graphs would be undesirably poor as the theoretical bound only suggests the worst-case performance in an expected manner. In other words, for graphs with smaller $\phi$, there exists more risk for worse performance in practice.
> > >
> > > The experimental results accord with our theory and intuitive understandings based on our method. Concretely, the homophily ratio of graphs in Table 4 is roughly positively correlated with the *absolute* performance in Table 1. For homophily graphs, with larger homophily ratio $\phi$, the achieved classification accuracy of the model is usually higher (using similar splits, $\phi$: Cora > Pubmed > Citeseer, $Acc$: Cora > Pubmed > Citeseer. Coauthor-Physics has the largest $\phi$ and as well the highest accuracy). On heterophilic graphs in Table 3, the accuracy scores are overall worse than those on homophilic ones in Table 1.
> > >
> > > > "Although the authors cite a few previous papers to justify this unexpected experimental result in Q2, the graph homophily ratio never serves as the assumption or guides the model design in those papers."
> > >
> > > We tend to have different interpretations on these cited works. In [R9], the graph homophily ratio does motivate their model design. In Fig.1 of [R9], the authors plot the classification accuracy of low-frequency signals, high-frequency signals and the proposed method, w.r.t. the probability of inter-connection $q$. Note that $q$ is the probability that two nodes from different classes are connected, a larger $q$ indicates a small $\phi$ while a smaller $q$ indicates a high $\phi$. Fig.1 of [R9] demonstrates that when the graph becomes more heterophilic ($\phi$ becomes larger), high-frequency information is more important for classification. This observation motivates the design of FAGNN, which utilizes both low-frequency and high-frequency information in a GNN framework.
> > >
> > > > "The working mechanism of GNN is to model the neighborhood distribution of different nodes instead of directly aligning the distance between the connected nodes. The authors are encouraged to rethink the assumption and rationality behind it."
> > >
> > > Thanks for raising the discussion on this point which can help us to further clarify the message and the principle of how our model works. We agree that GNN models can model the neighborhood distribution by iteratively aggregating the embeddings from the neighborhoods. In fact, our learning objective essentially plays a similar role. To be specific, while from a **local** view, our adopted loss function aims at maximizing the similarity of two connected nodes, the embedding of a target node would be influenced by all its neighboring nodes from a **global** perspective. As a result, the learning signal for a target node is decided by the distribution of its neighbored nodes. As long as the distribution of neighbored nodes for target nodes from different classes are distinguishable (which is easy to meet in the studied heterophily graphs [R10]), our model can learn desired representations for downstream classification.
> > >
> > > As further justification, we next demonstrate that even if the graph homophily ratio is very small (i.e., zero), our method can still yield reasonable prediction as long as different classes of nodes exhibit distinct neighborhood distributions and feature distributions. As a proof-of-concept, we consider a 4-class classification task, and the four classes are denoted by A,B,C,D, respectively. For a node from class A, 50\% of its neighboring nodes are from class B, and the remaining 50\% are from class D. For a node from class B, 50\% of its neighboring nodes are from class C, and the remaining 50\% are from class A. Then similar for nodes from class C and D. In this case, the graph homophily ratio is $\phi = 0$ as there are no intra-class edges, but nodes from different classes have distinct neighborhood distributions, which still enables the model to learn distinguishable embeddings.
> > >
> > > References
> > >
> > > [R9] Bo, Deyu, et al. "Beyond low-frequency information in graph convolutional networks." Proceedings of the AAAI Conference on Artificial Intelligence. Vol. 35. No. 5. 2021.
> > >
> > > [R10] Yao Ma, Xiaorui Liu, Neil Shah, and Jiliang Tang. Is homophily a necessity for graph neural networks? In ICLR, 2022.

---

### Official Review · Reviewer_Nj2N · 2022-10-24

**Confidence:** 3
**Clarity, Quality, Novelty And Reproducibility:** see above
**Correctness:** 4
**Technical Novelty And Significance:** 4
**Empirical Novelty And Significance:** 3
**Recommendation:** 8

**Strength And Weaknesses:**

Disclaimer: I am familiar with contrastive learning but am unfamiliar with the literature on graph representation learning / graph contrastive learning. So I cannot judge on the novelty compared to the existing literature. That said, the literature overview and the benchmarks shown in this paper look legit convincing, and the method does appear SOTA. Both novel ideas (for positive and negative pairs) are interesting. Therefore I believe this is a strong paper and a strong accept.

I do not have any major issues. A number of things were not entirely clear to me, but it should be easy to edit them for clarity.

* Section 2.3: it is not immediately clear how this section refers to the rest. I only understood it several pages later. Perhaps one introductory sentence could be added to explain the purpose of this section.

* Equation 2 -- the InfoNCE loss usually has positive pair in the denominator as well, not only in the numerator (see e.g. SimCLR paper). Here you don't include the positive pair in the denominator. Why is that? Please comment.

* Section 3: it was not entirely clear to me here what is the input to the network, and what are the i's in z_i. Later I understood that every node in the graph has an associated feature vector, and these features constitute the input neurons of the network. This is what you mean by "node-level", but I think it may be helpful to spell it out more clearly.

* page 5: "st" -> "set"

* Theorem 1 sounds like a really strong and important result, but inspection of Appendix A.1 suggests that it is a rather straightforward consequence of the results from two other papers. I think it would be prudent to cite both of them in the main text and to write something like "The proof is a direct consequence of the results of \citet{} and \citet{}".

* Section 5.1: It would be great to have Table 4 in the main text but I understand that the space constraint may not allow it. But I would suggest to give some summary stats in the text of this section, perhaps mention that the number of nodes varied from 2k to 170k, etc. Maybe also give the range of the feature dimensionalities, etc.

* Section 5.1: "Implemention details" paragraph could mention the network architecture. Was it a fully-connected network (MLP)? Something else?

* Figure 2: I would suggest to connect all GRACE points with a solid green line.

* Section 5.4: In SimCLR, the number of negative examples is determined by the mini-batch size. Is the same true in GRACE?

* Related: I am actually not sure what if you used mini-batches for Local-GCL. What was the mini-batch size? Or did you use full-batch training instead of stochastic gradient descent? Your description on page 5 sounds as if you use *all* graph nodes as negative pairs, meaning that your M in Equation 2 is equal to the number of nodes. Is my understanding correct? If so, then does this mean that you did full-batch training? Otherwise how did you implement mini-batches?

* Equation 8 holds for a non-graph contrastive learning models such as SimCLR. Does it mean that one could you user random Fourier features approach in SimCLR and related non-graph methods?

* Section B.1 and also the main text: As I said, I am unfamiliar with the graph contrastive learning literature, but I am surprised to read that most existing methods usg some elaborate data augmentation schemes. Your approach (take graph edges as positive pairs) seems much simpler. Why has it not been adopted before?

**Summary Of The Paper:**

The paper "Localized Graph Contrastive Learning" suggests a contrastive learning method (Local-GCL) for graph data, based on the InfoNCE loss. In this setting, each graph node is associated with a feature vector, and the goal is to learn a meaningful representation of the graph nodes (representation quality can be measured using a linear classifier for known node labels). Local-GCL is based on two ideas: (1) use graph edges instead of data augmentations for positive pairs; (2) use random Fourier features to quickly compute negative part of the loss. The authors show that Local-GCL achieves SOTA performance on several benchmark datasets.

**Summary Of The Review:**

The paper suggests several novel ideas to graph contrastive learning and achieves SOTA results. Strong accept. However, disclaimer: I am unfamiliar with the literature on graph contrastive learning.

---

> ### Author Response · Authors · 2022-11-16
> **Responses to Reviewer Nj2N, batch 1 of 2**
>
> We thank the reviewer for the thorough review. We notice that your questions are mainly about some unclear expressions. We are pleased to answer the reviewer's questions one by one.
>
> Q1: Add an introductory sentence to explain Section 2.3.
>
> A1: We thank the reviewer for noticing that it requires an introductory sentence at the beginning of Sec. 2.3. In the revised version, we have highlighted the scalability issue of InfoNCE loss when applied to large graphs, which motivates the utilization of Random Features.
>
> Q2: Equation 2: why the positive pair is not included in the denominator?
>
> A2: This is in fact a typo. The positive pair should be covered in the denominator together with negative ones so that the pairs in the denominator are randomly sampled from the product of two marginal distributions (then it can be related to mutual information). Note that in Local-GCL's loss (Eq. 5), the positive term is also included in the denominator. We thank the reviewer for noticing this mistake and we've revised it in the latest version.
>
> Q3: Sec3. Not clear what is the input to the network, and what is $i$ in $z_i$.
>
> A3: We thank the reviewer for mentioning the potential vagueness that could help us to improve the presentation. Just as you comprehend, the input is node features (together with the graph structure), and the output is each node's embedding $z_i$. The encoder could be either an MLP or a GNN. We've added a short description in the first paragraph of Sec.3 so that readers that are not familiar with this area can quickly understand the framework of node-level contrastive learning.
>
> Q4: Theorem 1.
>
> A4: As suggested, we've added the citations, which clearly suggest that our theorem can be easily derived from the conclusions of these two works.
>
> Q5: Add descriptional sentences about datasets in the main text.
>
> A5: Due to space limitations, we cannot add more descriptions about the dataset statistics in Sec 5.1, but we stated that ogbn-arxiv has about 17k nodes, and on it a lot of previous methods can not scale.
>
> Q6: Implementation details about the encoder architecture.
>
> A6: In implementation details, we have already mentioned that the encoder is a two-layer GCN by default.
>
> Q7: Questions about mini-batch training and GRACE.
>
> A7: In SimCLR for image representation learning, the number of negative examples is (about double) of the batch size. However, in GRACE (one can view it as a graph version of SimCLR), the number of negative examples is $2|\mathcal{V} -1|$, where $\mathcal{V}$ is the number of nodes in the graph. So GRACE does not use mini-batch training, and this is exactly the reason that it cannot scale to large graphs. (We've revised Fig. 2 as suggested.)
>
> We didn't use mini-batch training for Local-GCL. Instead, at each training epoch, we forward the model to get all node embeddings within the graph and optimize the model using our Local-GCL loss (as the reviewer understands, we use all graph nodes as negative pairs).
>
> Q8: Can Random Fourier Features be applied to non-graph contrastive learning scenarios?
>
> A8: We agree that RFF can be applied to non-graph contrastive learning tasks such as SimCLR in CV. However, there is an essential difference between graph (node) and images, which makes it unnecessary to approximate the computation of SimCLR loss in CV: the main bottleneck in image contrastive learning is the backbone model, e.g., ResNet-18 or ResNet-50. This prevents a large batch size from being used. For example, in SimCLR, the largest batch size can only be 8192, and this is based on the fact that they use over 32 TPUs. For such a baseline, the number of negative examples is small, and the computation cost for the contrastive loss function is almost negligible compared with the backbone encoder. As a result, it is unnecessary to use RFF to accelerate in CV. By contrast, for node representation learning, it is not hard to train the model in a full-graph manner. In this case, the number of negative examples could be huge. As a result, RFF can benefit a lot in both time and memory complexity.

---

> > ### Comment · Reviewer_Nj2N · 2022-11-16
> > **Thank you**
> >
> > Thank you for the reply and for helpful clarifications. After reading the other reviews and your responses, I keep my score at 8.
> >
> > > we stated that ogbn-arxiv has about 17k nodes
> >
> > I think there is a typo here (also in the manuscript): it has about 170k nodes, right?
> >
> > > However, there is an essential difference between graph (node) and images, which makes it unnecessary to approximate the computation of SimCLR loss in CV: the main bottleneck in image contrastive learning is the backbone model, e.g., ResNet-18 or ResNet-50. This prevents a large batch size from being used. [...] By contrast, for node representation learning, it is not hard to train the model in a full-graph manner.
> >
> > I am not sure I correctly understood this point. Do you mean that ResNet models are so large that they take up GPU memory and therefore only relatively small batches (e.g. on the order of 1024) can be used (even if we were to use your Random Fourier Features trick)? Whereas the networks you use in GCL-Local are much smaller and therefore GPU memory is available for larger batches? Did I understand the argument correctly?
> >
> > If so, I imagine that at some graph size, full-batch in GCL-Local would become unfeasible. What if ogbn-arxiv had 10x more nodes? 100x more nodes? Would you then switch to minibatch learning? This is just a clarification question.

---

> > > ### Author Response · Authors · 2022-11-16
> > > **We thank the reviewer for the timely reply**
> > >
> > > Size of ogbn-arxiv: it should be 170k nodes.
> > >
> > > For CL in CV: you are correct. The large-scale vision models take most of the memory, so only small batch sizes can be used. For graphs, we are able to use full-batch training as GNN models are shallow and small-scale.
> > >
> > > For graphs with millions of nodes, it is hard to fit the whole graph in the GPU for training, so mini-batch training and neighborhood sampling is necessary. In this case, we have to sample a set of target nodes, then create a subgraph covering the k-hop neighborhood of the target nodes (k is GNN's depth). Then GNN is applied to the subgraph to get the embeddings of the target nodes as well as their neighbors.

---

> > ### Comment · Reviewer_Nj2N · 2022-11-16
> > **Two further questions**
> >
> > 1. You refer to homophilic and heterophilic graphs, but do not show the value of \phi (homophily ratio) for the graphs that you are working with. Would it be possible to add \phi column to Table 4?
> >
> > 2. According to the Table 5, you used GCN encoder for 5 graphs and MLP encoder for 2 other graphs. Why? Would it be possible to use GCN or MLP everywhere? How much worse the results would be then?

---

> > > ### Author Response · Authors · 2022-11-17
> > > **Further updates**
> > >
> > > We thank the reviewer for listing the homophily ratio of each dataset in Table 4. We've updated it in the latest version.
> > >
> > > For the encoders actually we can implement the encoder with any model that takes node features and/or graph structures as input to generate node embedding. However, in the context of self-supervised learning on graphs, only shallow GCNs and same-sized MLPs are considered for fair comparison. As GCN can incorporate the structural information and node features for learning node representations, GCN as the encoder can usually give a higher performance on most datasets compared with MLP. Motivated by the previous SOTA method CCA-SSG [A] which demonstrates better performance on CS when using MLP instead of GCN (in Sec. 5.1), in our experiments we also try replace the GCN model with the same-sized MLP.
> > >
> > > |        |   Cora  | Citeseer | Pubmed | Computer | Photo| CS | Physics
> > > |  ----  | ----  |  ----  |---- |---- |---- |---- |---- |
> > > | MLP | 78.3 $\pm$ 0.5 | 70.3 $\pm$ 0.6 |79.6 $\pm$ 0.5| 82.40 $\pm$ 0.51| 90.87 $\pm$ 0.36| 94.90 $\pm$ 0.19| 96.33 $\pm$ 0.13|
> > > | GCN | 84.5 $\pm$ 0.4 | 73.6 $\pm$ 0.4 |82.1 $\pm$ 0.5|88.81 $\pm$ 0.37| 93.25 $\pm$ 0.40| 92.96 $\pm$ 0.26| 95.45 $\pm$ 0.11|
> > >
> > > As demonstrated in the table, we empirically find that LocalGCL with MLP as the encoder performs much better than GCN no CS and Physicis, and the accuracy is much higher than previous SOTAs. Note that preivous SOTA CCA-SSG also uses MLP on CS, but the performance (93.31) is only marginally better than LocalGCL with GCN, and much worse than Local-GCL with MLP. We also test CCA-SSG with MLP encoder on Physics but it gives lower result than GCN. The reason behind this phenomenon might be that Local-GCL's objective function can implictly inject the graph structure's knowledge into the encoder's parameters. For CS and Physics, the dimension of input features is very high (6,805 and 8,451) so the node feature contains much more information than the structures. As a result, an MLP model optmizied with structure-regularized loss can better utilize the rich node features than GCN which pays more information to graph structures. Our conclusion is, when using GCN as the encoder everywhere, the proposed Local-GCL can still demonstrate the best performacne on most datasets, and compeitive performance on the remaining, but if we switch to an MLP encoder on the feature-rich dataset, our method can perform much better than others (even if they also use an MLP encoder).
> > >
> > > References
> > >
> > > [A] Hengrui Zhang, Qitian Wu, Junchi Yan, David Wipf, and Philip S Yu. From canonical correlation analysis to self-supervised graph neural networks. In NeurIPS, 2021.

---

> ### Author Response · Authors · 2022-11-16
> **Responses to Reviewer Nj2N, batch 2 of 2**
>
> Q9: Why wasn't the idea studied in previous works?
>
> A9: This is an interesting question. Actually, utilizing the graph structure information  for unsupervised node representation learning itself is not a novel idea. For example, the classical network embedding method, LINE[1], just treats first-order neighbors as positive examples and uses randomly sampled nodes as negative examples. The difference, to be rough, is just the loss function, where that of LINE is a cross-entropy one, while ours is based on InfoNCE. In our opinion, the concept of contrastive learning for unsupervised representation learning was first proposed in late 2018 (CPC[2]), and became popular until the beginning of 2020 (SimCLR[5]), in computer vision for learning image representations. As in images, there isn't structure information available for constructing positive pairs, data augmentations are indispensable. Then, motivated by the success of contrastive (and some non-contrastive) methods in CV, some researchers tried to transfer them to the graph domain (e.g., DIM[2] and DGI[4], SimCLR[5] and GRACE[6], BYOL[7] and BGRL[8], BarlowTwins[9] and Graph BarlowTwins[10]). As they are direct applications of these methods from CV to graph, they simply design (or borrow existing) graph data augmentation strategies without considering the unique property of the graph (e.g., connections between nodes). In this paper, we rethink the necessity of data augmentations in graph contrastive learning, and we demonstrate that with simple connecting nodes as positive examples, we can achieve competitive or even better performance than the augmentation-based methods.
>
>
> References:
>
> [1] Tang, Jian, et al. "Line: Large-scale information network embedding." Proceedings of the 24th international conference on world wide web. 2015.
>
> [2] Oord, Aaron van den, Yazhe Li, and Oriol Vinyals. "Representation learning with contrastive predictive coding." arXiv preprint arXiv:1807.03748 (2018).
>
> [3] Hjelm, R. Devon, et al. "Learning deep representations by mutual information estimation and maximization." arXiv preprint arXiv:1808.06670 (2018).
>
> [4] Petar Velickovic, William Fedus, William L. Hamilton, Pietro Liò, Yoshua Bengio, and R. Devon Hjelm. Deep graph infomax. In ICLR, 2019.
>
> [5] Ting Chen, Simon Kornblith, Mohammad Norouzi, and Geoffrey E. Hinton. A simple framework for contrastive learning of visual representations. In ICML, Proceedings of Machine Learning Research, pages 1597–1607, 2020.
>
> [6] Yanqiao Zhu, Yichen Xu, Feng Yu, Qiang Liu, Shu Wu, and Liang Wang. Deep graph contrastive representation learning. arXiv preprint arXiv:2006.04131, 2020.
>
> [7] Grill, Jean-Bastien, et al. "Bootstrap your own latent-a new approach to self-supervised learning." Advances in neural information processing systems 33 (2020): 21271-21284.
>
> [8] Shantanu Thakoor, Corentin Tallec, Mohammad Gheshlaghi Azar, Rémi Munos, Petar Velickovic, and Michal Valko. Bootstrapped representation learning on graphs. arXiv preprint arXiv:2102.06514, 2021.
>
> [9] Jure Zbontar, Li Jing, Ishan Misra, Yann LeCun, and Stéphane Deny. Barlow twins: Self=supervised learning via redundancy reduction. In ICML, 2021.
>
> [10] Bielak, Piotr, Tomasz Kajdanowicz, and Nitesh V. Chawla. "Graph Barlow Twins: A self-supervised representation learning framework for graphs." Knowledge-Based Systems 256 (2022): 109631.

---

### Official Review · Reviewer_4oqi · 2022-10-25

**Confidence:** 4
**Correctness:** 3
**Technical Novelty And Significance:** 3
**Empirical Novelty And Significance:** 3
**Recommendation:** 6

**Clarity, Quality, Novelty And Reproducibility:**

Clarity: The method and background are explained well. The implementation details and experimental setting are also clear.

Quality: The theoretical analysis is sound, and provides support for the adopted method. The experiments are convincing and appear to be well-executed, demonstrating the superior computational efficiency of the proposed method.

Novelty: Although the idea is simple and efficient, overall the work is lacking novelty, as discussed above.

Reproducibility: The code and the shell file that can produce the results presented in paper are provided. Assuming these are made public, the reproducibility is sufficient.


**Details Of Ethics Concerns:**

Not applicable.

**Strength And Weaknesses:**

Strengths:
S1. The use of first order neighbourhoods for positive sampling is theoretically justified under the homophily assumption.
S2. The proposed method is computationally efficient and experiments show that the approach can scale to larger graphs.
S3. The paper is well organized and explains the approach very clearly.

Weaknesses:

1. The idea relies on homophily, however the performance improvement for heterophilic graphs (table 4) is more pronounced. It is not clear how this is possible and the paper provides no explanation apart from unsupported conjectures. The result tends to undermine the other experimental results for the homophilic, drawing into question whether the observed good performance is genuine and whether it arises for the reasons posed in the paper.

2. Although the proposed approach is effective, the idea of using neighbours as positive samples in graph contrastive learning has been proposed before, albeit recently. The use of kernel approximations random Fourier features has been proposed in other (non-graph) self-supervised learning work. This diminishes the novelty and impact of the paper.


Questions and Comments

1.	Novelty: The idea of using the neighbour as a positive sample in contrastive learning has been proposed before (for example, in [R1, R2]). [R2] was only published in July, so may be consider concurrent work, but [R1] appeared in February. Although the definition of a “neighbour” in [R1] is more complicated (making it much less computationally efficient than the proposed technique), this does diminish the novelty of the proposed approach.

If it is accepted that graph contrastive learning using neighbours as positive samples is a known idea, then it’s not clear to me that the paper makes a sufficient theoretical or experimental contribution. Each theorem, while interesting, is derived fairly straightforwardly from existing results. The experiments are conducted over multiple datasets and compare against numerous baselines, but the results for the graphs displaying heterophily are unexplained and a concern. The kernel approximations to make the computation efficient are a valuable contribution, although in the context of non-graph learning, the use of random Fourier features has been proposed in [R3]. The proposal to use structured orthogonal random features in this context is novel, to the best of my knowledge, and does result in a significant computational saving.

Can the authors discuss the related work that I have identified, stressing the differences, and reinforcing the novelty of the contribution?

2. Can the authors provide a better explanation for the performance in the heterophilic graph case?

3. The paper does not report the results of any statistical significance tests. I would encourage the reporting of (bootstrapped) confidence intervals rather than 1 standard deviation. Single split results also give a false impression of the variability of the performance. Wilcoxon pair tests between the best and second-best techniques would provide more compelling evidence that there is a genuine performance difference.

[R1] Zihan Lin, Changxin Tian, Yupeng Hou, Wayne Xin Zhao, “Improving Graph Collaborative Filtering with Neighborhood-enriched Contrastive Learning,” in Proceedings of the ACM Web Conference (WWW), 2022.

[R2] Sun, Z., Harit, A., Cristea, A. I., Yu, J., Shi, L., & Al Moubayed, N. “Contrastive Learning with Heterogeneous Graph Attention Networks on Short Text Classification,” In Proc. International Joint Conference on Neural Networks (IJCNN), 2022.

[R3] Li, Y., Pogodin, R., Sutherland, D. J., & Gretton, A. (2021). Self-supervised learning with kernel dependence maximization. Advances in Neural Information Processing Systems, 34, 15543-15556.


**Summary Of The Paper:**

This work proposes a contrastive learning method for self-supervised node-level tasks in graph domain. It involves (1) sampling positive samples from the first order neighbourhoods and (2) kernelizing negative loss to reduce the training time and memory overheads.
The main contribution of this paper is presenting a simple yet efficient and intuitively reasonable idea regarding the use of first level neighbourhoods for selecting positive samples and achieving competitive performance with state-of-the-art methods. Some theoretical results are presented to support the adopted approach.


**Summary Of The Review:**

The paper proposes a simple technique for self-supervised learning for node classification. Theoretical results are provided to motivate the approach. The paper introduces a kernelization strategy to significantly reduce the computational burden. Although simple, the experiments demonstrate that the method can achieve SOTA performance. The main misgiving concerning the paper is the novelty and the impact, considering that a similar approach has already been proposed in other work.

---

> ### Author Response · Authors · 2022-11-16
> **Responses to Reviewer 4oqi, batch 1 of 3**
>
> We thank the reviewer for spending time reading our paper and giving such thorough comments and valuable feedbacks. Here'd like to answer the reviewer's questions and address some concerns.
>
> **Empirical results**
>
> Q1: More explanations for the results on heterophilic graphs.
>
> Although a little bit counter-intuitive, the better performance of our method on heterophilic graphs does exist, and can be justified through the differences between the two distinguished methods of generating positive pairs: 1) data augmentations on graph data; 2) two connected nodes being a positive pair. Two recent studies [A] and [B] both demonstrate that existing data augmentations on graphs, such as graph diffusion used in MVGRL, edge dropping and feature masking used in GRACE, GCA, BGRL and CCA-SSG have larger impacts on the high-frequency information of the graph than the low-frequency information. As a result, the differences between two graph views are about low-frequency components. Considering that contrastive learning aims at maximizng the mutual information shared between the two views, only the invariant information (low-frequency) is encouraged to be learned by the embeddings, whle the middle and high frequency information is discarded.
>
> On the other hand, studies on GNNs for heterophilic graphs [C] demonstrate that for heterophilic graphs, the high-frequency information in the graph is more effective for downstream classification performance. Considering that data augmentation based methods focus on low-frequency information while neglecting high-frequency one, it is not strange that they cannot give satisfactory performance. By contrast, as high frequency knowledge in the graph represents the differences between the node feature with its neighborhood features, the edge-wise contrastive loss used in our work can better capture the differences in the information between two connected nodes, thus performing better on heterophilic graphs.
>
> We've added the detailed explanations for the performance on heterophilic graphs in the revised version.
>
> Referecnes:
>
> [A] Wang, Haonan, et al. "Augmentation-Free Graph Contrastive Learning." arXiv preprint arXiv:2204.04874 (2022).
>
> [B] Liu, Nian, et al. "Revisiting Graph Contrastive Learning from the Perspective of Graph Spectrum." arXiv preprint arXiv:2210.02330 (2022).
>
> [C] Bo, Deyu, et al. "Beyond low-frequency information in graph convolutional networks." Proceedings of the AAAI Conference on Artificial Intelligence. Vol. 35. No. 5. 2021.

---

> ### Author Response · Authors · 2022-11-16
> **Responses to Reviewer 4oqi, batch 2 of 3**
>
> **Novelty**
>
> Q2: Differences to existing neighborhood-based contrastive learning methods.
>
> A2: We thank the reviewer for suggesting these up-to-date works which share a similar idea of using neighboring nodes to construct positive examples. We didn't notice [R1] because it is particularly designed for recommender systems, and we didn't notice the second one because 1) it is a very recent paper, and 2) it does not imply that it uses neighbors in its title. Though the basic idea of these two works and ours is similar: neighboring nodes can serve as positive examples for contrastive learning, there are significant differences in the implementations.
>
> In [R1], Eq.(6), the positive pairs are constructed between user $u$'s embedding at layer $0$, and layer $k$. The motivation for such a design is that graph convolution (e.g, user $u$'s embedding at layer $k$) is a summary of the information of the neighboring nodes, so it contrasts a user's embeddings from different layers. The issue with such a design, however, is that embeddings from different layers lie in different spaces and thus cannot be theoretically justified using mutual information maximization theory. Besides, Eq.(6) in [R1] is a node-centric contrastive objective, while LocalGCL is an edge-centric one. As a result, [R1] is not guaranteed to recover the graph structure, while our LocalGCL can (see Proof of Theorem 1 in Appendix A.1, where the LocalGCL loss can recover the graph adjacency matrix $\tilde{A}$).
>
> [R2] is closer to our work because it also performs edge-centric contrastive learning. The major difference between Eq.(5) in [R2] and ours is that [R2] considers all nodes within the $k$-hop neighborhoods as positive examples. Though $k$ should be a very important hyperparameter that can significantly affect the model's performance, [R2] fails to give any analysis or empirical results (e.g., ablation studies on $k$ ) on how to decide it. Besides, as shown in III.D in [R2], the authors seem to choose a large $k$, which can be computationally inefficient. We agree that utilizing the graph structure information for unsupervised node representation learning itself is not a novel idea. For example, the classic network embedding method, LINE[R4], just treats first-order neighbors as positive examples and uses randomly sampled nodes as negative examples. However, no previous work has successfully applied such a simple idea to node-level contrastive representation learning. The major contribution of this work is that, we demonstrate that contrastive learning with purely first-order neighbors, without high-order connectivity and without data augmentation, can achieve competitive or even better performance than those with high-order connectivity or data augmentations.
>
> References:
>
> [R1] Zihan Lin, Changxin Tian, Yupeng Hou, Wayne Xin Zhao, “Improving Graph Collaborative Filtering with Neighborhood-enriched Contrastive Learning,” in Proceedings of the ACM Web Conference (WWW), 2022.
>
> [R2] Sun, Z., Harit, A., Cristea, A. I., Yu, J., Shi, L., \& Al Moubayed, N. “Contrastive Learning with Heterogeneous Graph Attention Networks on Short Text Classification,” In Proc. International Joint Conference on Neural Networks (IJCNN), 2022.
>
> [R4] Tang, Jian, et al. "Line: Large-scale information network embedding." Proceedings of the 24th international conference on world wide web. 2015.

---

> ### Author Response · Authors · 2022-11-16
> **Responses to Reviewer 4oqi, batch 3 of 3**
>
> Q3: Differences to another method using kernel approximation.
>
> A3: Though [R3] also adopts kernel approximation for self-supervised learning, [R3] and our work largely differ in both motivation and implementation. In [R3], the authors propose to maximize the HSIC between two transformed views instead of mutual information (which is usually realized using InfoNCE loss). As the computation of HSIC directly requires a gram matrix of the kernel function, it naturally employs RFF to reduce the complexity of the computation of HSIC. From this perspective, SSL-HSIC is a replacement of the InfoNCE-loss-based contrastive learning methods. Although performing similarly, they are two distinct methods with different motivations. By contrast, there isn't an explicit formulation of the kernel function in InfoNCE loss. In this paper, we show that the InfoNCE loss could be transformed into a kernelized version and thus can be accelerated using kernel approximation.
>
> References:
>
> [R3] Li, Y., Pogodin, R., Sutherland, D. J., \& Gretton, A. (2021). Self-supervised learning with kernel dependence maximization. Advances in Neural Information Processing Systems, 34, 15543-15556.
>
> Q4: Results of statistical significance tests.
>
> A4: We agree with the reviewer that simply reporting the standard deviation might not be strong enough. However, to our knowledge, in the context of self-supervised learning on graphs, it is a common practice to report the averaged score with a standard deviation [A,B,C,D]. In order for consistency with these baselines, we simply report the performance in terms of mean accuracy with the standard deviation of 20 random trials. For Cora, Citeseer and PubMed, we follow the common practice using the fixed public split as nearly all the important baseline models are based on this split. This is the same for ogbn-arxiv dataset, where the public split is recommended by the dataset contributor. Note that for Computer, Photo, CS, and Physics, we use random split so there should not be concerns about the model's robustness against dataset split.
>
> References:
>
> [R5] Yanqiao Zhu, Yichen Xu, Feng Yu, Qiang Liu, Shu Wu, and Liang Wang. Deep graph contrastive
> representation learning. arXiv preprint arXiv:2006.04131, 2020b.
>
> [R6] Kaveh Hassani and Amir Hosein Khas Ahmadi. Contrastive multi-view representation learning on
> graphs. In ICML, 2020.
>
> [R7] Shantanu Thakoor, Corentin Tallec, Mohammad Gheshlaghi Azar, Rémi Munos, Petar Velickovic, and
> Michal Valko. Large-scale representation learning on graphs via bootstrapping. ICLR, 2022.
>
> [R8] Hengrui Zhang, Qitian Wu, Junchi Yan, David Wipf, and Philip S Yu. From canonical correlation
> analysis to self-supervised graph neural networks. In NeurIPS, 2021.

---

> > ### Comment · Reviewer_4oqi · 2022-11-17
> > **Acknowledgment of author response**
> >
> > I appreciate the thoroughness of the response provided by the authors.
> > (1) The discussion of the related work has certainly helped to identify and clarify the novel contribution and the important differences in the approaches.
> >
> > (2) While I agree that the argument as to why the technique achieves substantial outperformance for the heterophilic examples is probably reasonable, and I think the discussion has improved, I still find this to be a weaker portion of the paper. These type of results should have motivated further experimentation and analysis to provide support for the conjectures offered to explain the results. For example, an analysis of the local neighbourhoods could identify to what extent the one-hop neighbourhoods of nodes from the same class are similar. The proximity of the constructed representations could have been investigated and the extent to which this correlated with neighbourhood similarity could have been assessed. I understand that it is unreasonable to expect further experimentation during the rebuttal phase, but if the research  endeavour is continued, I would encourage the authors to delve further into this issue. In some ways, this is the most impressive performance improvement in the paper, but the experimental analysis stops with the reporting of accuracy.
> >
> > (3) I don't think it is reasonable to argue that it is not worthwhile performing tests for statistical significance because other papers have not conducted such an analysis. Consistency with existing work is not a good argument in this case. I think there is consensus in the machine learning and statistics research communities that experimental analysis is incomplete if there is not an adequate assessment of the experimental uncertainty and the significance of the findings (via confidence intervals or statistical comparison tests or preferably both). Reporting standard deviations is not really sufficient for an adequate assessment of the statistical significance of the results.

---

> > > ### Author Response · Authors · 2022-11-18
> > > **Thanks for providing more feedbacks**
> > >
> > > We thank the reviewer again for providing more feedbacks.
> > >
> > > For more empirical justification for heterophily graphs, we are sorry that we are unable to conduct more experiments to justify the assumptions for heterophily graphs. We agree with the reviewer that it is beneficial and meaningful to do further study on the differences between the patterns that previous methods and our method can learn, and what kind of information different models actually focus. We are willing to do more research on this issue subsequently.
> > >
> > > We are sorry but we didn't mean it is not worthwhile to do the statistical significance tests. We agree with the reviewer that it is insufficient to simply report mean performance with standard deviations. As a result, we've added the results of bootstrapped confidence interval (with 95\% confidence) of our method and the best baseline model CCA-SSG in Appendix C.5. The results are based on 20 random initializations.
> > >
> > > bootstrapped Confidence Interval (with 95\% confidence)
> > >
> > > |       |   Cora  | Citeseer | Pubmed | Computer | Photo| CS | Physics
> > > |  ----  | ----  |  ----  |---- |---- |---- |---- |---- |
> > > | CCA-SSG | [84.06, 84.30]| [73.05, 73.21] | [81.51, 81.72]| [88.57, 88.86] | [93.10, 93.22] | [93.25, 93.41] | [95.37, 95.41]|
> > > | Local-GCL | [84.43, 84.75] | [73.45, 73.83] | [81.82, 82.29] | [88.66, 89.06 ] | [93.20, 93.48] | [94.79, 94.98] | [96.26, 96.35] |
> > >
> > > We also conduct Wilcoxon signed-rank test on the proposed Local-GCL and the second-best model CCA-SSG, using the averaged test accuracy on the 7 datasets in Table 1. The p-value of CCA-SSG and Local-GCL's performance on the 7 datasets is 0.0325 < 0.05. As a result, we can conclude that there is a genuine performance difference.
> > >
> > > We thank the reviewer again for the valuable suggestions.

---

### Official Review · Reviewer_q5an · 2022-10-25

**Confidence:** 4
**Correctness:** 2
**Technical Novelty And Significance:** 2
**Empirical Novelty And Significance:** 2
**Recommendation:** 5

**Clarity, Quality, Novelty And Reproducibility:**

I think the paper brings some interesting thoughts and should be quite reproducible, considering the code+instructions that will be provided in supplementary material.

In terms of clarity, there are a few issues due to mistakes in equations / definitions which I believe it would be important to address (in addition to typos, which only marginally reduce the understanding of the document):

- Lemma 1 calls $\Sigma$ the eigenvalues matrix $\Lambda$.
- Lemma 2 refers to $\hat{y} = X \rightarrow [r]$ while it should be $\hat{y} = V \rightarrow [r]$
- Theorem 1: the dimensionality of $B^*$ is $k \times c$, while it should be $d \times c$
- the RFF map is defined as being $\phi : \R^d \rightarrow \R^D$ but, while the $\omega_i$ samples are just $D$, the dimensionality of $\phi$ in Eq 1 seems $2D$ instead.
- finally, I think using $\phi$ to refer both to homophily ratio and to the RFF map might lead to some confusion


Some additional issues relate to the soundness of the theoretical presentation. In my opinion, a few concepts require a better explaination that also takes into consideration assumptions and boundary conditions. For instance:

- The simplification of Eq 9 that allows us to precalculate all the $\phi(h_j)$ is possible because $\phi(h_i)$ is fixed and can be taken out of the sum. This, I guess, is the reason why the negative loss is calculated on all nodes and not just the ones which do not belong to a given node's neighborhood. While the impact of this choice does not seem to be too important, I think it should be made explicit and thoroughly discussed (i.e. why it makes sense, when it might not work, how the error changes wrt the average degree vs graph size).

- The proof of Theorem 1 relies on a set of assumptions which to me are not completely clear:
	- Lemma 1 builds above Theorem 5 from Balestriero and LeCun (2022) which specifically refers to the global minimiser of the SimCLR loss. How is Local-GCL's loss, which includes a negative loss component which is approximated, related to it? Is there a proof of its compatibility with the one of SimCLR?

	- Lemma 2 builds above Theorem B.3 from HaoChen et al. (2021) which explicitly introduces eigenvalues $\lambda_1, \dots, \lambda_{k+1}$ as the smallest ones, while in the main body of the paper (just before Definition 1) they are defined as being in a descending order so the bigger $i$ is the smaller the eigenvalue (i.e. the first ones are the larger ones)

	- Correlation between dimensionality and performances is introduces without saying anything about boundary conditions. Nothing is said about what the maximum dimensionality is, and how it relates not just to the size of the graph but also its properties. For instance, if we have $k$ connected components, we cannot have $d > |V| - k$ because the smallest k eigenvalues would all be zero


In terms of novelty (and in the light of the previous issues) my impression is that, while the paper brings an interesting solution derived from previous works, in absence of a sound theory behind it the theoretical contribution would just look incremental.

On the experimental side:

- it is not clear which of the two (orthogonal) model improvements contributes the most to  performances: intuitively, looking at GRACE's performances, I'd say it is the possibility of using all negative pairs in a more scalable way; getting positive pairs from the 1-hop neighbors, on the other hand, mainly allows us to get rid of augmentations. To verify this, I think it would be useful to have an ablation experiment showing how a model such as GRACE with approximate neg loss would perform. It looks like the main difference in GRACE is the presence of the intra-view negative pairs term, which I think could be computed as efficiently as the inter-view one.

- In Table 1, the difference of the top two means in the Photo dataset is not really significant given the values of means and stds and the amount of experiments. While this does not make a difference in practice (I think the claim "our approach is competitive with the SOA while being more scalable" is correct), it would be fair to make both of the values bold as it cannot be statistically determined whether one is actually better than the other.

- after the discussion on how positive pairs are generated, results on heterophilic graphs might come unexpected but, as the authors correctly say citing Ma et Al's work, this can happen e.g. due to the model learning to recognize different neigbor distributions. This claim, however, is unsupported by experiments which would allow us both to verify the reason of this behavior and learn (e.g. from counterexamples) when this model would work and when it would not.


**Strength And Weaknesses:**

Strengths:
- the method proposes a method to significantly reduce the computational complexity of the calculation of negative loss in GCL models
- the experiments show results which are promising, especially when compared to other models' running times

Weaknesses:
- the paper is not always clear due to mistakes or ambiguous definitions
- I have some doubts about the contribution and the soundness of the theoretical part of the paper (see below)
- the experimental section still leaves some open questions (related to the contribution brought by the two different components and the performances on heterophilic graphs)

**Summary Of The Paper:**

The paper introduces Local-GCL, a graph contrastive learning approach that tackles two of the most common issues of GCL models: the need for augmentation to generate positive pairs (which is non-trivial in the graph-learning setting) and the need for a high amount of negative pairs comparisons (whose quadratic complexity makes the problem intractable for large graphs). It does so by, respectively, creating positive pairs from first-order neighbors, and introducing an approximated contrastive loss computation that approximates the original loss with a much lower complexity (linear instead of quadratic). The authors provide theoretical justifications for their model and show empirically that it performs competitively to the state of the art while having a much smaller computational footprint.

**Summary Of The Review:**

The paper aims at improving CGL by providing a model which is, at the same time, independent of graph augmentations and more scalable. Motivations are clear and relevant and the presentation of the related work is in my opinion sufficient to place this work in the proper context.

While empirical results are in my opinion quite positive, the paper still leaves some open questions about the contributions brought individually by the two components implemented in the model. I also think some claims should be better supported, so that the theoretical contribution becomes more rigorous and less incremental.

---

> ### Author Response · Authors · 2022-11-16
> **Responses to Reviewer q5an, batch 1 of 3.**
>
> We thank the reviewer for the thorough comments and constructive suggestions, which will be a great help for us to improve our work. We are pleased to address your concerns here:
>
> **Clarity**
>
> Q1: Typos that affect reading.
>
> A1: We've fixed these typos and unclear equations/definitions in the revised version.
>
> Q2: Unclear assumptions or boundary conditions in the proof of Theorem 1:
>
> A2: For the compatibility of approximated Local-GCL's loss and the original InfoNCE loss, we didn't cover this discussion in the paper because RFF is just making numerical approximations to reduce the computational complexity without affecting the optimal solution of the loss. The proof is trivial: according to Lemma 3 in Appendix A.2, the Gaussian kernel function $e^{-\Vert \mathbf{x} - \mathbf{y} \Vert_2^2 / 2\tau}$ can be unbiasedly approximated with RFF. As a result, Eq. (9) is also an unbiased approximation of the denominator of Eq. (5). As we didn't approximate the positive terms of Eq. (5), the approximated Local-GCL loss should be an unbiased estimation of Eq.(5). As a result, when the surrogate loss function reaches its global minimum, Eq. (5) also reaches its global minimum.
>
> For Lemma 2 (eigenvalues), we are sorry for the typo that incorrectly describes the order of eigenvalues. Actually, it should be an **ascending order** so that $\lambda_1$ is the smallest one, and $\lambda_{|\mathcal{V}|}$ is the largest one. This is consistent with the analysis below Eq.(7), that the larger the embedding dimension $d$ is (the larger $\lambda_d$ is), the smaller the linear classification error bound can be. We've fixed this typo in the revised version.
>
>
> For the boundary conditions, we do assume that the maximum dimension $d$ should be no larger than the number of nodes in the graph $|\mathcal{V}|$ (see the last paragraph before the definition of graph homophily ratio). Considering that most studied graphs have at least tens of thousands of nodes, which is much larger than the embedding dimension actually used, this assumption is reasonable.  For the number of connected components $k$, you might want to mean $d < k$ because in Theorem 1 $\lambda_d$ is d-th smallest eigenvalue, and in this case, $\lambda_{d+1}$ = 0. However, even in this extreme case, Theorem 1 is still correct as the error will be bounded by infinity when $\lambda_{d+1} = 0$.

---

> ### Author Response · Authors · 2022-11-16
> **Responses to Reviewer q5an, batch 2 of 3.**
>
> **Concerns about experimental results**
>
> Q3: Which of the two model improvements contributes to the performance?
>
> A3: We would like to emphasize that the utilization approximation of the negative loss is just to reduce the computational complexity, and it will not bring performance improvement. All the performance improvements are actually brought by the positive example of construction using neighbors. To better demonstrate this, we further provide the results of directly using Eq.(5) as the objective function in the following table (we add this table in the revised version):
>
>
> |         | Cora   | Citeseer  | Pubmed |
> |  ----   | ----- | ---- | ----  |
> | w/o approx.     | 84.5 $\pm$ 0.4 | 73.6 $\pm$ 0.4 | 82.2 $\pm$ 0.5|
> | Local-GCL(RFF)  | 84.3 $\pm$ 0.5 | 73.5 $\pm$ 0.7 | 81.9 $\pm$ 0.7|
> | Local-GCL(SORF) | 84.5 $\pm$ 0.4 | 73.6 $\pm$ 0.5 | 82.1 $\pm$ 0.5|
>
> In this table, w/o approx denotes the performance when directly using Eq.(5) as the loss function. The results above demonstrate that using approximated loss not only brings no improvements but may also cause a slight performance drop. This is because there is randomness in the approximation, which can only be reduced instead of eliminated. Besides, SORF shows better performance than RFF, which is because SORF has a much smaller approximation variance than RFF, when using the same projection dimension.
>
> Q4: Highlight the strongest baseline in Photo dataset.
>
> A4: We thank the reviewer for this suggestion. AFGRL's mean performance on Photo is close to ours but has a smaller deviation. We have highlighted their results as well in the revised version as suggested.
>
> Q5: Explanations for results on heterophilic graphs.
>
> A5: Although a little bit counter-intuitive, the better performance of our method on heterophilic graphs does exist, and can be justified through the differences between the two distinguished methods of generating positive pairs: 1) data augmentations on graph data; 2) two connected nodes being a positive pair. Two recent studies [4] and [5] both demonstrate that existing data augmentations on graphs, such as graph diffusion used in MVGRL, edge dropping and feature masking used in GRACE, GCA, BGRL and CCA-SSG have larger impacts on the high-frequency information of the graph than the low-frequency information. As a result, the differences between two graph views are about low-frequency components. Considering that contrastive learning aims at maximizng the mutual information shared between the two views, only the invariant information (low-frequency) is encouraged to be learned by the embeddings, whle the middle and high frequency information is discarded.
>
> On the other hand, studies on GNNs for heterophilic graphs [6] demonstrate that for heterophilic graphs, the high-frequency information in the graph is more effective for downstream classification performance. Considering that data augmentation based methods focus on low-frequency information while neglecting high-frequency one, it is not strange that they cannot give satisfactory performance. By contrast, as high frequency knowledge in the graph represents the differences between the node feature with its neighborhood features, the edge-wise contrastive loss used in our work can better capture the differences in the information between two connected nodes, thus performing better on heterophilic graphs.
>
> We've added the detailed explanations for the performance on heterophilic graphs in the revised version.
>
> Referecnes:
>
> [4] Wang, Haonan, et al. "Augmentation-Free Graph Contrastive Learning." arXiv preprint arXiv:2204.04874 (2022).
>
> [5] Liu, Nian, et al. "Revisiting Graph Contrastive Learning from the Perspective of Graph Spectrum." arXiv preprint arXiv:2210.02330 (2022).
>
> [6] Bo, Deyu, et al. "Beyond low-frequency information in graph convolutional networks." Proceedings of the AAAI Conference on Artificial Intelligence. Vol. 35. No. 5. 2021.

---

> ### Author Response · Authors · 2022-11-16
> **Responses to Reviewer q5an, batch 3 of 3.**
>
> **Soundness of the theoretical presentation**
>
> Q6: Whether to cover positive terms in the denominator of the objective function.
>
> A6: This is an interesting and important question which does worth discussing. First, let's review the InfoNCE loss, and the NT-Xent loss used in SimCLR. Given embeddings of two views: $\{(\mathbf{z}^A_i, \mathbf{z}^B_i)\}_{i=1}^N$, and consider the loss for view A (for view B is just symmetric):
>
> InfoNCE:
> $$
> \mathcal{L}_{InfoNCE} = -\sum\limits\_{i=1}^N \log \frac{\exp(\mathbf{z}_i^A\cdot \mathbf{z}_i^B / \tau)}{\sum\limits\_{j=1}^N \exp(\mathbf{z}_i^A\cdot \mathbf{z}_j^B / \tau)}
> $$
>
> NT-Xent in SimCLR:
> $$
> \mathcal{L}_{SimCLR} = -\sum\limits\_{i=1}^N \log \frac{\exp(\mathbf{z}_i^A\cdot \mathbf{z}_i^B / \tau)}{\sum\limits\_{j = 1}^N \exp(\mathbf{z}_i^A\cdot \mathbf{z}_j^B / \tau) + \sum\limits\_{j \neq i} \exp(\mathbf{z}_i^A\cdot \mathbf{z}_j^A / \tau)}.
> $$
>
> In both the original InfoNCE loss and NT-Xent loss in SimCLR, the positive loss is covered in the denominator together with the negative ones. This is not a casual operation but is due to its theoretical connection to the Mutual Information maximization principle. In the CPC paper, the authors show that the InfoNCE loss is a lower bound of mutual information. Concretely, denote the random variables of two views' embeddings by $Z^A$ and $Z^B$ respectively, the InfoNCE tries to maximize the Mutual Information between $Z^A$ and $Z^B$, i.e., $I(Z^A, Z^B)$, or the KL-Divergence between two distributions $p(Z^A, Z^B)$, $p(Z^A)p(Z^B)$. From this perspective, the numerator term in InfoNCE does represent a positive example, while the denominator term means two embeddings randomly and uniformly sampled from two marginal distributions, so it is necessary to cover a positive term together with true negative ones in the denominator, and this is the reason that the negative loss is calculated on all nodes in Eq.9, although we believe removing the positive loss from the denominator will not make significant differences to the model performance.
>
> Besides, it is easy to implement if we would like to cover only nodes that do not belong to a given node's neighborhood. As noted in the paper, for a target node $i$, we compute its positive loss by:
> $$
>     \mathcal{L}\_{\rm pos}(i) = -\log \sum\limits\_{j \in \mathcal{N}(i)} \exp (\mathbf{z}_i^{\top}\mathbf{z}_j / \tau) / |\mathcal{N}(i)|,
> $$
>
> and original the negative loss is:
>
> $$
>      \mathcal{L}\_{\rm neg}(i) = \log \sum\limits\_{k \in \mathcal{V}} \exp (\mathbf{z}_i^{\top}\mathbf{z}_k / \tau).
> $$
>
> If we only consider nodes do not belong to the given node's neighborhood:
>
> $$ \mathcal{L}'_{\text{neg}}(i) = \log \sum\limits\_{k \notin \mathcal{N}(i)} \exp (\mathbf{z}_i^{\top}\mathbf{z}_k / \tau) $$
>
> As the $\log$ operation is applied at the last step, we neglect it, and we can find that:
>
> $$\sum\limits\_{k \notin \mathcal{N}(i)} \exp (\mathbf{z}_i^{\top}\mathbf{z}_k / \tau)  = \sum\limits\_{k \in \mathcal{V}} \exp (\mathbf{z}_i^{\top}\mathbf{z}_k / \tau) - \frac{\sum\_{j \in \mathcal{N}(i)}\exp (\mathbf{z}_i^{\top}\mathbf{z}_j / \tau)}{|\mathcal{N}(i)|} \cdot {|\mathcal{N}(i)|}$$
>
> As a result, the new negative term can be directly derived from the original negative term and positive term without additional computational cost. Based on this, we further conduct experiments using the new negative loss that excludes neighboring nodes. We compare the results in the table below (we use exactly the same set of hyperparameters and results are averaged over 20 random trials):
>
> |         |  Cora   | Citeseer | Pubmed |
> |  ----   | ----  |  ----  |----  |
> | LocalGCL(origin)   | 84.5 $\pm$ 0.4 | 73.6 $\pm$ 0.4 |82.1 $\pm$ 0.5|
> | LocalGCL(new)     |  84.6 $\pm$ 0.4 | 73.5 $\pm$ 0.5 |81.9 $\pm$ 0.5|
>
> The results are within our expectations: excluding the positive terms from the denominator  makes little difference to the model performance. Though on some datasets, it can lead to marginal improvement, it might hurt the performance a little for others. Besides, this result might also indicate that forcing the embeddings to distribute uniformly in the latent space may be more beneficial than simply separating negative pairs [1].
>
> The reviewer also mentions that it would be better to discuss the error changes of two implementations w.r.t. the average degree v.s. graph size. We agree with the reviewer that this will be an interesting topic. However, considering that in the studied datasets, the average (even the largest) node degree is much smaller than the total number of nodes, it is likely that we cannot get a meaningful result. A more appropriate study case should be on synthetic datasets where we can control the graph size and avg. node degrees, but it is out of the range of this paper.
>
> References:
>
> [1] Wang, Tongzhou, and Phillip Isola. "Understanding contrastive representation learning through alignment and uniformity on the hypersphere." International Conference on Machine Learning. PMLR, 2020.

---

### Public Comment · ~Benedek_Andras_Rozemberczki1 · 2022-11-05
**Misattribution of datasets**

The paper misattributed the authorship of the Chameleons and Squirrels datasets. These datasets were proposed in this ICLR submission:

https://openreview.net/forum?id=HJxiMAVtPH

The Pei et al. paper cited by the authors took the Squirrel and Chameleons datasets and used those for benchmarking, but had nothing to do with the creation of the datasets. The correct citation for the paper which proposed the datasets is:

```bibtex
>@article{musae,
          author = {Rozemberczki, Benedek and Allen, Carl and Sarkar, Rik},
          title = {{Multi-Scale Attributed Node Embedding}},
          journal = {Journal of Complex Networks},
          volume = {9},
          number = {2},
          year = {2021},
}
```

---

> ### Author Response · Authors · 2022-11-05
> **Thanks for giving the correct citation.**
>
> We thank the commenter for giving the correct citation of the Squirrel and Chameleon datasets. We will fix it in the revised version!

---

### Author Response · Authors · 2022-11-16
**General responses to all reviewers.**

We thank all the reviewers for their thorough comments and valuable opinions/suggestions. We've given detailed responses to each reviewer, and we've also uploaded the revised version of this paper.  The major differences of the revised paper from the initial version are:

1) We fixed some typos and unclear expressions that might affect reading.

2) We add detailed explanations for the results on heterophilc graphs, based on the effect of data augmentations on the different frequency components of the graph information.

3) We extend LOCAL-GCL with other designs of positive example construction, negative
example selection, and self-supervised objective functions to evaluate the effectiveness of each single
component of the proposed method. The new results highlight the flexibility and effectivenss of combining the proposed methods with other techniques to get a more advanced model.

We hope that our responses and the revised paper can help answer the reviewers' questions and address the concerns.

---

### Author Response · Authors · 2022-12-08
**A Summary of Discussions and Clarification from Authors**

Dear (Senior) Area Chairs and Reviewers,

We appreciate your time for reviewing this paper and the provided constructive feedback. After the discussions with reviewers, we noticed that there still exists some (potential) divergent opinions on our contributions of performance improvement over existing self-supervised learning methods on graphs.

- Reviewer q5an, 4oqi and Nj2N appreciated our contributions: the problem is well motivated (q5an, 4oqi, Nj2N), the method is novel (Nj2N), and the experimental results achieve the SOTA performance (4oqi, Nj2N).
- Reviewer BqrR holds some reservations and raised a point that our improvement is not significant over the self-supervised learning methods without using negative pairs, e.g., BGRL [1] and CCA-SSG [2].

While we have responded to the reviewer BqrR, there is no feedback that suggests whether our message is received. To resolve some potentially existing yet undeclared concerns, we'd like to clarify two points about our work to facilitate the reviewing process towards a precise evaluation of our work.

**About improvements over self-supervised learning methods without negative pairs.** In fact, our single model Local-GCL has achieved SOTA performance across 9 benchmark datasets in Table 1 and 3, and critically the best runner-ups (on some datasets are non-contrastive methods while on others are contrastive ones) are different among datasets. And as a by-product, we also conduct extra experiments in Table 7, which shows that our designs can also further boost the accuracy and efficiency of non-contrastive methods (BGRL and CCA-SSG). We believe these results strongly support our claims.

**The motivation of studying contrastive learning with negative pairs.** One natural question is whether it is meaningful to improve the efficiency of computing the negative term (for models using negative pairs) since there already exists models without negative pairs achieving satisfactory performance. Our answer is YES since contrastive learning using negative pairs is still the mainstream class of self-supervised approaches in either graph learning or general vision tasks. First, on the empirical side, using negative pairs or not is an 'equal choice' and there is no evidence suggesting that one algorithm can win over the other in the majority of self-supervised learning cases [3]. Second, on the theoretical side, using negative pairs can be more informative and principled, which makes the model more reasonable and grounded [4,5,6,7]. Third, from a broader perspective, studying how to perform contrastive on graph data more effectively is still an open problem that is underexplored in terms of more informative positive examples [8,9], hard negative and false negative examples [10,11], how to improve the objective function [12,13], etc. Based on these facts, our research problem is well-motivated, of practical significance, and could benefit the community.

Best,

Paper 2283 authors

References:

[1] Shantanu Thakoor, Corentin Tallec, Mohammad Gheshlaghi Azar, Rémi Munos, Petar Velickovic, and Michal Valko. Bootstrapped representation learning on graphs. arXiv preprint arXiv:2102.06514, 2021.

[2] Hengrui Zhang, Qitian Wu, Junchi Yan, David Wipf, and Philip S Yu. From canonical correlation analysis to self-supervised graph neural networks. In NeurIPS, 2021.

[3] Chen, Xinlei, Saining Xie, and Kaiming He. An empirical study of training self-supervised vision transformers. In CVPR, 2021.

[4] Oord, Aaron van den, Yazhe Li, and Oriol Vinyals. Representation learning with contrastive predictive coding. arXiv preprint arXiv:1807.03748 (2018).

[5] Poole, Ben, et al. On variational bounds of mutual information. International Conference on Machine Learning. PMLR, 2019.

[6] Tosh, Christopher, Akshay Krishnamurthy, and Daniel Hsu. Contrastive learning, multi-view redundancy, and linear models. Algorithmic Learning Theory. PMLR, 2021.

[7] Pokle, Ashwini, et al. Contrasting the landscape of contrastive and non-contrastive learning. In AISTATS, 2022.

[8] Han, Tengda, Weidi Xie, and Andrew Zisserman. Self-supervised co-training for video representation learning. In NeurIPS, 2020.

[9] Dwibedi, Debidatta, et al. With a little help from my friends: Nearest-neighbor contrastive learning of visual representations. In CVPR, 2021.

[10] Kalantidis, Yannis, Mert Bulent Sariyildiz, Noe Pion, Philippe Weinzaepfel, and Diane Larlus. Hard negative mixing for contrastive learning. In NeurIPS, 2020.

[11] Huynh, Tri, et al. Boosting contrastive self-supervised learning with false negative cancellation. In WACV, 2022.

[12] Lux, Florian, Ching-Yi Chen, and Ngoc Thang Vu. Combining Contrastive and Non-Contrastive Losses for Fine-Tuning Pretrained Models in Speech Analysis. In IEEE SLT, 2022.

[13] Ziyang Liu and Hao Feng and Chaokun Wang. Rethinking Temperature in Graph Contrastive Learning. Open Review, 2022.

---

### Decision · Program_Chairs · 2023-01-20

**Decision:**

Reject

**Justification For Why Not Higher Score:**

This paper does not reach the bar of ICLR, which lacks clear motivations and technical novelty. Two contributions on the contractiveness with neighbors and acceleration are not self-standing. More details can be found in my meta review.

I had several round discussions with the authors in the public thread and noticed the authors set the research problem is a very narrow scope. Moreover, the authors continued repeating their claims but failed to convince me.

**Justification For Why Not Lower Score:**

N/A

**Metareview: Summary, Strengths And Weaknesses:**

Based on the collected information from all of reviewers and my personal judgement, I can make the recommendation on this paper, **rejection**. Here are the comments that I summarized, which include my opinion and evidence.

**Research Problem**

The authors consider the well-defined graph contrastive learning problem in the node classification setting.

**Motivation**

In this paper, there are two major motivations. (1) Prevailing contrastive learning methods rely on predefined augmentation techniques and (2) their quadratic computational complexity might lead to inconsistency and inefficiency problems. I believe the authors also notice there is a branch of data augment-free category in contrastive learning, where the authors compare with them in the experimental part. However, there is no discussion on this point in the introduction part, which makes the first point not very strong.

Another thing I would like to point out is that the above two motivations are not coherent, although they focus on the positive/negative samples in the contrastive learning. They might be two separate papers or each of them are not strong enough to be a single paper.

**Philosophy**

To handle the above two challenges, the authors aim to fabricate the positive examples for each node directly using its first order neighbor and devise a kernelized contrastive loss, which could be approximately computed in linear time and space complexity with respect to the graph size.
For the first point, beyond several key references provided Reviewer 4oqi, there is another a missing key reference, Graph-MLP [1], which contains a neighboring contrastive. The idea is almost the same. For the second point, I also agree with Reviewer 4oqi that each theorem, while interesting, is derived fairly and straightforwardly from existing results. Based on those, the novelty of this paper is limited when taking the sufficient literature into consideration.

**Negative-Free Method**

Reviewer BqrR raised a point on the negative-free category, which is also a crucial branch in contrastive learning. Ref [2,3] demonstrate they can achieve even better performance without negative samples. If we consider the negative-free methods, the second motivation is not strong, either.

**Homophily**

Two reviewers have concerns on the performance on homophily and heterophilic graphs. Although the authors provide the extra explanation, I am still confused on this part. Some experimental results might provide better understanding. For example, checking the performance of other competitive methods on homophily and heterophilic graphs.

**Technical Part**

I talk a lot in the philosophy part. So, I skip here.

**Experiments**

(1) From the results in Table 1, the proposed method can achieve significant improvements on *CS* and *Physics* datasets, compared with the second-best method.

(2) The running time comparison is incomplete, which only involves four algorithms. Moreover, there is no significant improvement in terms of efficiency. The negative-free methods are not included, either.

(3) The numbers of competitive methods in Tables 1, 2 & 3 are inconsistent.

(4) It is good to see the performance of the proposed method on the large dataset (Table 2).

Three reviewers support the rejection recommendation.

[1] Hu, Y., You, H., Wang, Z., Wang, Z., Zhou, E., & Gao, Y. (2021). Graph-MLP: node classification without message passing in graph. arXiv preprint arXiv:2106.04051.
[2] Chen, X., & He, K. (2021). Exploring simple siamese representation learning. In Proceedings of the IEEE/CVF Conference on Computer Vision and Pattern Recognition (pp. 15750-15758).
[3] Grill, J. B., Strub, F., Altché, F., Tallec, C., Richemond, P., Buchatskaya, E., ... & Valko, M. (2020). Bootstrap your own latent-a new approach to self-supervised learning. Advances in neural information processing systems, 33, 21271-21284.

**Summary Of Ac-Reviewer Meeting:**

This is not a borderline paper.